# Penalty-based Methods for Simple Bilevel Optimization under Hölderian Error Bounds

**Pengyu Chen**[*]
School of Data Science
Fudan University
pychen22@m.fudan.edu.cn

**Xu Shi**[*]
School of Data Science
Fudan University
xshi22@m.fudan.edu.cn

**Rujun Jiang**[†]
School of Data Science
Fudan University
rjjiang@fudan.edu.cn

**Jiulin Wang**
School of Data Science
Fudan University
wangjiulin@fudan.edu.cn

## Abstract

This paper investigates simple bilevel optimization problems where we minimize an upper-level objective over the optimal solution set of a convex lower-level objective. Existing methods for such problems either only guarantee asymptotic convergence, have slow sublinear rates, or require strong assumptions. To address these challenges, we propose a penalization framework that delineates the relationship between approximate solutions of the original problem and its reformulated counterparts. This framework accommodates varying assumptions regarding smoothness and convexity, enabling the application of specific methods with different complexity results. Specifically, when both upper- and lower-level objectives are composite convex functions, under an $\alpha$-Hölderian error bound condition and certain mild assumptions, our algorithm attains an $(\epsilon, \epsilon^\beta)$-optimal solution of the original problem for any $\beta > 0$ within $\mathcal{O}\left(\sqrt{1/\epsilon^{\max\{\alpha,\beta\}}}\right)$ iterations. The result can be improved further if the smooth part of the upper-level objective is strongly convex. We also establish complexity results when the upper- and lower-level objectives are general nonsmooth functions. Numerical experiments demonstrate the effectiveness of our algorithms.

## 1 Introduction

Bilevel optimization involves embedding one optimization problem within another, creating a hierarchical structure where the upper-level problem's feasible set is influenced by the lower-level problem. This framework frequently occurs in various real-world scenarios, such as meta-learning [Bertinetto et al., 2018, Rajeswaran et al., 2019], hyper-parameter optimization [Chen et al., 2024, Franceschi et al., 2018, Shaban et al., 2019], reinforcement learning [Mingyi et al., 2020] and adversarial learning [Bishop et al., 2020, Wang et al., 2021, 2022]. In this paper, we concentrate on a subset of bilevel optimization known as simple bilevel optimization (SBO), which has garnered significant interest in the machine learning community due to its relevance in dictionary learning [Beck and Sabach, 2014, Jiang et al., 2023], lexicographic optimization [Kissel et al., 2020, Gong et al., 2021], lifelong learning [Malitsky, 2017, Jiang et al., 2023]; see more details in Appendix A.

---

[*]Equal contribution
[†]Corresponding author

38th Conference on Neural Information Processing Systems (NeurIPS 2024).

SBO aims to find an optimal solution that minimizes the upper-level objective over the solution set of the lower-level problem. In other words, we are interested in solving the following problem:

$$\min_{\mathbf{x}\in\mathbb{R}^n} F(\mathbf{x}) \quad \text{s.t.} \quad \mathbf{x} \in \arg\min_{\mathbf{z}\in\mathbb{R}^n} G(\mathbf{z}). \tag{P}$$

Here $F, G : \mathbb{R}^n \to \mathbb{R}\bigcup\{\infty\}$ are proper, convex, and lower semi-continuous functions. We also assume that the optimal solution set of the lower-level problem, denoted as $X_{\text{opt}}$, is nonempty. Moreover, since $G$ is convex and lower semi-continuous, it holds that $X_{\text{opt}}$ is closed and convex [Bertsekas et al., 2003, Proposition 1.2.2 and Page 49].

In this paper, we first reformulate problem (P) into the constrained form:

$$\min_{\mathbf{x}\in\mathbb{R}^n} F(\mathbf{x}) \quad \text{s.t.} \quad G(\mathbf{x}) - G^* \leq 0, \tag{$\text{P}_{\text{Val}}$}$$

where $G^*$ represents the optimal value of the unconstrained lower-level problem.

Based on this reformulation, we consider the following penalization of ($\text{P}_{\text{Val}}$),

$$\min_{\mathbf{x}\in\mathbb{R}^n} \Phi_\gamma(\mathbf{x}) = F(\mathbf{x}) + \gamma p(\mathbf{x}), \tag{$\text{P}_\gamma$}$$

where $p(\mathbf{x}) := G(\mathbf{x}) - G^*$ is the so-called residual function and $\gamma > 0$ is the penalized parameter. Obviously, we have $p(\mathbf{x}) \geq 0$, and $p(\mathbf{x}) = 0$ if and only if $\mathbf{x} \in X_{\text{opt}}$.

Denote $F^*$ and $G^*$ as the optimal values of problem (P) and the lower-level problem $\min_{\mathbf{x}\in\mathbb{R}^n} G(\mathbf{x})$, respectively. We aim to find an $(\epsilon_F, \epsilon_G)$-optimal solution $\tilde{\mathbf{x}}^*$ of problem (P), which satisfies

$$F(\tilde{\mathbf{x}}^*) - F^* \leq \epsilon_F, \quad G(\tilde{\mathbf{x}}^*) - G^* \leq \epsilon_G. \tag{1}$$

Moreover, a point $\tilde{\mathbf{x}}_\gamma^*$ is said to be an $\epsilon$-optimal solution of problem ($\text{P}_\gamma$) if

$$\Phi_\gamma(\tilde{\mathbf{x}}_\gamma^*) - \Phi_\gamma^* \leq \epsilon,$$

where $\Phi_\gamma^*$ is the optimal value of problem ($\text{P}_\gamma$).

## 1.1 Related work

Various approaches have been developed to solve problem (P) [Cabot, 2005, Solodov, 2007, Sabach and Shtern, 2017, Dutta and Pandit, 2020, Gong et al., 2021]. Among those, one category that is the most related to penalization formulation ($\text{P}_\gamma$) is the regularization method, which integrates the upper- and lower-level objectives through Tikhonov regularization [Tikhonov and Arsenin, 1977]

$$\min_{\mathbf{x}\in\mathbb{R}^n} \eta(\mathbf{x}) := \sigma F(\mathbf{x}) + G(\mathbf{x}), \tag{$\text{P}_{\text{Reg}}$}$$

where $\sigma$ is the so-called regularization parameter. When $F$ is strongly convex and its domain is compact, Amini and Yousefian [2019] extended the IR-PG method from Solodov [2007], which achieved a asymptotic convergence rate for the upper-level problem and a convergence rate of $\mathcal{O}\left(1/K^{0.5-b}\right)$ for the lower-level problem, where $b \in (0, 0.5)$. Malitsky [2017] studied a version of Tseng's accelerated gradient method and showed a convergence rate of $\mathcal{O}\left(1/K\right)$ for the lower-level problem, while the convergence rate for the upper-level objective is not explicitly provided. Kaushik and Yousefian [2021] proposed an iteratively regularized gradient (a-IRG) method which obtains a complexity of $\mathcal{O}\left(1/K^{0.5-b}\right)$ and $\mathcal{O}\left(1/K^b\right)$ for the upper- and lower-level objective, respectively, where $b \in (0, 0.5)$. Inspired by this research, and under a quasi-Lipschitz assumption for $F$, Merchav and Sabach [2023] introduced a bi-subgradient (Bi-SG) method. This method demonstrates convergence rates of $\mathcal{O}(1/K^b)$ and $\mathcal{O}(1/K^{1-b})$ for the lower- and upper-level objectives, respectively, where $b \in (0.5, 1)$. In their framework, the convergence rate of the upper-level objective can be improved to be linear when $F$ is strongly convex. Recently, under the weak-sharp minima assumption of the lower-level problem, Samadi et al. [2023] proposed a regularized accelerated proximal method (R-APM), showing a convergence rate of $\mathcal{O}(1/K^2)$ for both upper- and lower-level objectives. When the domain is compact and $F, G$ are both smooth, Giang-Tran et al. [2023] proposed an iteratively regularized conditional gradient (IR-CG) method, which ensures convergence rates of $\mathcal{O}(1/K^p)$ and $\mathcal{O}(1/K^{1-p})$ for upper- and lower-level objectives, respectively, where $p \in (0, 1)$.

Despite the abundance of existing methodologies yielding non-asymptotic convergence outcomes, their efficacy is frequently contingent upon additional assumptions. Denote $L_{f_1}$ and $L_{g_1}$ as the Lipschitz constants for the gradients of the smooth components in the upper- and lower-level objectives,

respectively. Specifically, when $F$ is strongly convex and $G$ is smooth, Beck and Sabach [2014] presented the Minimal Norm Gradient (MNG) method and provided the asymptotic convergence to the optimal solution set and a convergence rate of $\mathcal{O}\left(L_{g_1}^2/\epsilon^2\right)$ for the lower-level problem. When $F$ is assumed to be smooth, Jiang et al. [2023] introduced a conditional gradient-based bilevel optimization (CG-BiO) method, which invokes at most $\mathcal{O}\left(\max\{L_{f_1}/\epsilon_F, L_{g_1}/\epsilon_G\}\right)$ of linear optimization oracles to achieve an $(\epsilon_F, \epsilon_G)$-optimal solution. Shen et al. [2023] combined an online framework with the mirror descent algorithm and established a convergence rate of $\mathcal{O}(1/\epsilon^3)$ for both upper- and lower-level objectives, assuming a compact domain and boundedness of the functions and gradients at both levels. Furthermore, they showed that the convergence rate can be improved to $\mathcal{O}(1/\epsilon^2)$ under additional structural assumptions. For a concise overview of overall methodologies, including their assumptions and convergence outcomes, refer to Table 1 in Appendix B.

For general bilevel optimization problems, there have been recent results on convergent guarantees [Shen and Chen, 2023, Sow et al., 2022, Chen et al., 2023, Huang, 2023]. Among those, the one that is the most related to ours is [Shen and Chen, 2023]. It investigates the case when the upper-level objective is nonconvex and gives convergence results under additional assumptions [Shen and Chen, 2023, Theorem 3 and 4]. However, as the general bilevel optimization problem is nonconvex, the algorithms in the literature often converge to weak stationary points, while our method for SBO converges to global optimal solution.

## 1.2 Our approach

Our approach is straightforward. Firstly, we introduce a penalization framework delineating the connection between approximate solutions of problems (P) and (P$_\gamma$). This framework enables the attainment of an $(\epsilon_F, \epsilon_G)$-optimal solution by solving problem (P$_\gamma$) approximately. Subsequently, our focus shifts solely to resolving the unconstrained problem (P$_\gamma$). Depending on varying assumptions regarding smoothness and convexity, we can employ different methods such as the accelerated proximal gradient (APG) methods [Beck and Teboulle, 2009, Nesterov, 2013, Lin and Xiao, 2014] to solve problem (P$_\gamma$). We summarize our main contributions as follows.

- We propose a framework that explicitly examines the relationship between an $\epsilon$-optimal solution of penalty formulation (P$_\gamma$) and an $(\epsilon_F, \epsilon_G)$-optimal solution of problem (P). We also provide a lower bound for the metric $F(\mathbf{x}) - F^*$.

- When $F$ and $G$ are both composite convex functions, we provide a penalty-based APG algorithm that attains an $(\epsilon, \epsilon^\beta)$-optimal solution of problem (P) within $\mathcal{O}(\sqrt{1/\epsilon^{\max\{\alpha,\beta\}}})$ iterations. If the upper-level objective is strongly convex, the complexity can be improved to $\mathcal{O}(\sqrt{1/\epsilon^{\max\{\alpha-1,\beta-1\}}}\log\frac{1}{\epsilon})$. We also apply our method for the scenario where both the upper- and lower-level objectives are generalized nonsmooth convex functions.

- We present adaptive versions of PB-APG and PB-APG-sc with warm-start, which dynamically adjust the penalty parameters, and solve the associated penalized problem with adaptive accuracy. The adaptive ones have similar complexity results as their primal counterparts but can achieve superior performance in some experiments.

Utilizing the penalization method to address the original SBO problem is a novel approach. While Tikhonov regularization may seem similar to our framework, its principles differ. Implementing Tikhonov regularization necessitates the "slow condition" ($\lim_{k\to\infty}\sigma_k = 0, \sum_{k=0}^{\infty}\sigma_k = +\infty$), which requires iterative solutions for each iteration. In contrast, our method simply involves solving a single optimization problem (P$_\gamma$) for a given $\gamma$. Furthermore, we establish a relationship between the approximate solutions of the original bilevel problem and those of the reformulated single-level problem (P$_\gamma$) for a specific $\gamma$. This is the first theoretical result connecting the original bilevel problem to the penalization problem, accompanied by an optimal non-asymptotic complexity result.

## 2 The penalization framework

We begin by outlining specific assumptions for $F$ and $G$, as detailed below.

**Assumption 2.1.** The set $S := \bigcup_{\mathbf{x}\in X_{\text{opt}}}\partial F(\mathbf{x})$ is bounded with a diameter $l_F := \max_{\xi\in S}\|\xi\|$.

Note that the type of subdifferential $\partial F$ used here is the most general form for a convex function, as detailed in [Bertsekas et al., 2003, Section 4.2]. When the upper-level objective $F$ is non-convex,

we replace the assumption with the condition that the upper-level objective is Lipschitz continuous (cf. Theorems 2.7 and 2.8).

**Assumption 2.2** (Hölderian error bound). The function $p(\mathbf{x}) := G(\mathbf{x}) - G^*$ satisfies the Hölderian error bound with exponent $\alpha \geq 1$ and $\rho > 0$. Namely,

$$\text{dist}(\mathbf{x}, X_{\text{opt}})^\alpha \leq \rho p(\mathbf{x}), \forall \mathbf{x} \in \text{dom}(G),$$

where $\text{dist}(\mathbf{x}, X_{\text{opt}}) := \inf_{\mathbf{y} \in X_{\text{opt}}} \|\mathbf{x} - \mathbf{y}\|$.

We remark that Hölderian error bounds are satisfied by many practical problems and widely used in optimization literature [Pang, 1997, Bolte et al., 2017, Zhou and So, 2017, Roulet and d'Aspremont, 2020, Jiang and Li, 2022]. There are two notable special cases: (i) when $\alpha = 1$, we often refer to $X_{\text{opt}}$ as a set of weak sharp minima of $G$ [Burke and Ferris, 1993, Studniarski and Ward, 1999, Burke and Deng, 2005, Samadi et al., 2023]; (ii) when $\alpha = 2$, Assumption 2.2 is known as the quadratic growth condition [Drusvyatskiy and Lewis, 2018a]. Additional examples of functions exhibiting Hölderian error bound, along with their corresponding parameters, are presented in Appendix C.

We are now ready to establish the connection between approximate solutions of problems (P) and (P$_\gamma$). The subsequent two lemmas build upon the work of Shen and Chen [2023] for (general) bilevel optimization. Compared with their work, we generalize the exponent $\alpha$ from 2 to $\alpha \geq 1$, providing a more general result. Furthermore, we also derive a lower bound for the penalized parameter for all $\alpha \geq 1$ and present a theoretical framework for these scenarios.

**Lemma 2.3.** *Suppose that Assumptions 2.1 and 2.2 hold with $\alpha > 1$. Then, for any $\epsilon > 0$, an optimal solution of problem* (P) *is an $\epsilon$-optimal solution of problem* (P$_\gamma$) *when $\gamma \geq \rho l_F^\alpha (\alpha - 1)^{\alpha - 1} \alpha^{-\alpha} \epsilon^{1-\alpha}$.*

Lemma 2.3 establishes the relationship between an optimal solution of problem (P) and an $\epsilon$-optimal solution of problem (P$_\gamma$) when $\alpha > 1$. It also provides a lower bound for $\gamma$, which plays a pivotal role in the complexity results. The proofs of this paper are deferred to Appendix E.

The lemma presented below yields a more favorable outcome when $\alpha = 1$, which is referred to as exact penalization. Notably, this specific result is not discussed in Shen and Chen [2023].

**Lemma 2.4.** *Suppose that Assumptions 2.1 and 2.2 hold with $\alpha = 1$. Then an optimal solution of problem* (P) *is also an optimal solution of problem* (P$_\gamma$) *if $\gamma \geq \rho l_F$, and vice versa if $\gamma > \rho l_F$. In this case, we say that there is an exact penalization between problems* (P) *and* (P$_\gamma$).

For simplicity, we define

$$\gamma^* = \begin{cases} \rho l_F^\alpha (\alpha - 1)^{\alpha - 1} \alpha^{-\alpha} \epsilon^{1-\alpha} & \text{if } \alpha > 1 \\ \rho l_F & \text{if } \alpha = 1 \end{cases}. \qquad (2)$$

Based on Lemmas 2.3 and 2.4, we give an overall relationship of approximate solutions between problems (P$_\gamma$) and (P).

**Theorem 2.5.** *Suppose that Assumptions 2.1 and 2.2 hold. For any given $\epsilon > 0$ and $\beta > 0$, let*

$$\gamma = \gamma^* + \begin{cases} 2l_F^\beta \epsilon^{1-\beta} & \text{if } \alpha > 1, \\ l_F^\beta \epsilon^{1-\beta} & \text{if } \alpha = 1, \end{cases}$$

*with $\gamma^*$ defined in* (2). *If $\tilde{\mathbf{x}}_\gamma^*$ is an $\epsilon$-optimal solution of problem* (P$_\gamma$), *then $\tilde{\mathbf{x}}_\gamma^*$ is an $(\epsilon, l_F^{-\beta} \epsilon^\beta)$-optimal solution of problem* (P).

Particularly, we are also able to establish a lower bound for $F(\tilde{x}_\gamma^*) - F^*$ under the same conditions outlined in Theorem 2.5.

**Theorem 2.6.** *Suppose that the conditions in Theorem 2.5 hold. Then, $\tilde{\mathbf{x}}_\gamma^*$ satisfies the following suboptimality lower bound,*

$$F(\tilde{\mathbf{x}}_\gamma^*) - F^* \geq -l_F (\rho l_F^{-\beta} \epsilon^\beta)^{\frac{1}{\alpha}}.$$

By setting $\beta = \alpha$, we obtain $F(\tilde{\mathbf{x}}_\gamma^*) - F^* \geq -\rho^{\frac{1}{\alpha}} \epsilon$. which along with Theorem 2.5 gives

$$|F(\tilde{\mathbf{x}}_\gamma^*) - F^*| \leq \max\{\epsilon, \rho^{\frac{1}{\alpha}} \epsilon\}.$$

We emphasize that the lower bound established in Theorem 2.6 is an intrinsic property of problem (P) under Assumptions 2.1 and 2.2. This property is independent of the algorithms we present.

## 2.1 The upper-level function is non-convex

Note that the upper-level objective $F$ is required to be convex in the above context (cf. Theorem 2.5). This raises a question: while Theorem 2.5 establishes the relationship between approximate solutions of problems (P) and (P$_\gamma$), the distinction between the global or local optimal solutions of problem (P) and (P$_\gamma$) remains unclear when $F$ is non-convex.

We first establish the relationship between global optimal solutions of problems (P) and (P$_\gamma$) when $F$ is non-convex, which is similar to Theorem 2.5.

**Theorem 2.7.** *Suppose that Assumption 2.2 holds, $G$ is convex, and $F$ is $l$-Lipschitz continuous on* $\mathrm{dom}(F)$. *For any given $\epsilon > 0$ and $\beta > 0$, let*

$$\gamma = \gamma^* + \begin{cases} 2l^\beta \epsilon^{1-\beta} & \text{if } \alpha > 1, \\ l^\beta \epsilon^{1-\beta} & \text{if } \alpha = 1, \end{cases} \tag{3}$$

*where $\gamma^*$ is given by* (2). *If $\tilde{\mathbf{x}}_\gamma^*$ is an $\epsilon$-global optimal solution of problem* (P$_\gamma$)*, then $\tilde{\mathbf{x}}_\gamma^*$ is an* $(\epsilon, l^{-\beta}\epsilon^\beta)$-*global optimal solution of problem* (P).

Theorem 2.7 provides the relationship between the global optimal solutions of problems (P$_\gamma$) and (P). However, the relationship between local optimal solutions of these problems is more intricate than those of the global ones [Shen and Chen, 2023]. Given $r > 0$ and $\mathbf{z} \in \mathbb{R}^n$, define $\mathcal{B}(\mathbf{z}, r) := \{\mathbf{x} \in \mathbb{R}^n : \|\mathbf{x} - \mathbf{z}\| \leq r\}$. We present the following theorem, which demonstrates that local optimal solutions of problem (P$_\gamma$) can serve as approximate local optimal solutions of problem (P).

**Theorem 2.8.** *Suppose that Assumption 2.2 holds and $G$ is convex. Let $\mathbf{x}_\gamma^*$ be a local optimal solution of problem* (P$_\gamma$) *on $\mathcal{B}(\mathbf{x}_\gamma^*, r)$. Assume $F$ is $l$-Lipschitz continuous on $\mathcal{B}(\mathbf{x}_\gamma^*, r)$. Then $\mathbf{x}_\gamma^*$ is an approximate local optimal solution of problem* (P) *that satisfies $F(\mathbf{x}_\gamma^*) - F_\mathcal{B}^* \leq 0$ and $G(\mathbf{x}_\gamma^*) - G^* \leq \epsilon$ when $\alpha > 1$ and $\gamma \geq (\frac{\rho l^\alpha}{\epsilon^{\alpha-1}})^{\frac{1}{\alpha}}$, where $F_\mathcal{B}^*$ is the optimal value of problem* (P) *on $\mathcal{B}(\mathbf{x}_\gamma^*, r) \bigcap X_{\mathrm{opt}}$. Furthermore, $\mathbf{x}_\gamma^*$ is a local optimal solution of problem* (P) *when $\alpha = 1$ and $\gamma > \rho l$.*

Indeed, the relationship between approximate local optimal solutions of problems (P$_\gamma$) and (P) is more intricate than the connection among global solutions presented in Theorem 2.5. These interactions will be the focus of our future work. The proofs of Theorems 2.7 and 2.8 are presented in Appendixes E.5 and E.6.

## 3 Main algorithms

In this section, we concentrate on addressing problem (P), making various assumptions, and offering distinct convergence outcomes.

### 3.1 Both objectives are convex composite functions

In this scenario, we address problem (P) where $F$ and $G$ are both composite functions, i.e., $F = f_1 + f_2$ and $G = g_1 + g_2$.

**Assumption 3.1.** $F$ and $G$ satisfy the following assumptions.

(1) The gradient of $f_1(\mathbf{x})$, denoted as $\nabla f_1$, is $L_{f_1}$-Lipschitz continuous on $\mathrm{dom}(F)$;

(2) The gradient of $g_1(\mathbf{x})$, denoted as $\nabla g_1$, is $L_{g_1}$-Lipschitz continuous on $\mathrm{dom}(G)$;

(3) $f_2$ and $g_2$ are proper, convex, lower semicontinuous, and possibly non-smooth.

We remark that Assumption 3.1(1)(3) is more general than many existing papers in the literature. Specifically, while previous works such as Beck and Sabach [2014], Amini and Yousefian [2019], Jiang et al. [2023], Giang-Tran et al. [2023] require the upper-level objective to be smooth or strongly convex, we simply assume that $F$ is a composite function composed of a smooth convex function and a possibly non-smooth convex function. For the lower-level objective, previous works such as Beck and Sabach [2014], Amini and Yousefian [2019], Jiang et al. [2023], Giang-Tran et al. [2023] impose smoothness assumptions and, in some cases, convexity and compactness constraints on the domain; while our approach does not require these additional constraints, allowing for more flexibility and generality as presented in Assumption 3.1(2)(3).

We are now prepared to introduce two algorithms: the penalty-based accelerated proximal gradient (PB-APG) algorithm and its adaptive counterpart, the aPB-APG to solve problem ($P_\gamma$) and, subsequently, to obtain an $(\epsilon_F, \epsilon_G)$-optimal solution of problem (P).

To simplify notations, we omit the constant term $-\gamma G^*$, and rewrite problem ($P_\gamma$) as follows,

$$\min_{\mathbf{x} \in \mathbb{R}^n} \Phi_\gamma(\mathbf{x}) := \phi_\gamma(\mathbf{x}) + \psi_\gamma(\mathbf{x}), \tag{$P_\Phi$}$$

where $\phi_\gamma(\mathbf{x}) = f_1(\mathbf{x}) + \gamma g_1(\mathbf{x})$ and $\psi_\gamma(\mathbf{x}) = f_2(\mathbf{x}) + \gamma g_2(\mathbf{x})$ represent the smooth and nonsmooth parts, respectively. Then, it follows that the gradient of $\phi_\gamma(\mathbf{x})$ is $L_\gamma$-Lipschitz continuous with $L_\gamma = L_{f_1} + \gamma L_{g_1}$.

To implement the APG methods, we need another assumption concerning $\psi_\gamma(\mathbf{x})$.

**Assumption 3.2.** For any $\gamma > 0$, the function $\psi_\gamma(\mathbf{x})$ is prox-friendly, i.e., the proximal mapping

$$\operatorname{prox}_{t\psi_\gamma}(\mathbf{y}) := \arg\min_{\mathbf{x} \in \mathbb{R}^n}\{\psi_\gamma(\mathbf{x}) + \frac{1}{2t}\|\mathbf{x} - \mathbf{y}\|^2\},$$

is easy to compute for any $t > 0$.

The function $\psi_\gamma(\mathbf{x})$ represents the sum of two non-smooth functions, and proximal mapping for such function sums is widely studied and used in the literature [Yu, 2013, Pustelnik and Condat, 2017, Adly et al., 2019, Boob et al., 2023, Latafat et al., 2023]. This assumption is also a more general requirement compared to many existing algorithms [Sabach and Shtern, 2017, Giang-Tran et al., 2023]. It is important to note that our assumption is more general than existing literature. In the simple bilevel literature, when employing proximal mappings, researchers often consider the scenario where only one level contains a nonsmooth term (see, e.g., [Jiang et al., 2023, Doron and Shtern, 2023, Samadi et al., 2023, Merchav and Sabach, 2023]). In this case, the proximal mapping of the sum $f_2 + \gamma g_2$ is then reduced to the proximal mapping of either $f_2$ or $g_2$, which is a more easily satisfied condition.

### 3.1.1 Accelerated proximal gradient-based algorithm

We apply the APG algorithm [Beck and Teboulle, 2009, Lin and Xiao, 2014, Nesterov, 2013] to solve problem ($P_\Phi$), as outlined in Algorithm 1. Moreover, if the Lipschitz constant $L_\gamma$ is unknown or computationally infeasible, line search [Beck and Teboulle, 2009] can be adopted and will yield almost the same complexity bound. For brevity, we denote Algorithm 1 as $\hat{\mathbf{x}} = $ PB-APG$(\phi_\gamma, \psi_\gamma, L_{f_1}, L_{g_1}, \mathbf{x}_0, \epsilon)$, where $\hat{\mathbf{x}}$ represents an $\epsilon$-optimal solution of ($P_\Phi$).

---

**Algorithm 1** Penalty-based APG (PB-APG)

---
1: **Input:** $\gamma, L_\gamma = L_{f_1} + \gamma L_{g_1}, \mathbf{x}_{-1} = \mathbf{x}_0 \in \mathbb{R}^n, R > 0, t_{-1} = t_0 = 1, k = 0, \epsilon > 0$ and $\{t_k\}$.
2: **for** $k \geq 0$ **do**
3:      $\mathbf{y}_k = \mathbf{x}_k + t_k \left(t_{k-1}^{-1} - 1\right)(\mathbf{x}_k - \mathbf{x}_{k-1})$
4:      $\mathbf{x}_{k+1} = \operatorname{prox}_{L_\gamma^{-1}\psi_\gamma}(\mathbf{y}_k - L_\gamma^{-1}\nabla\phi_\gamma(\mathbf{y}_k))$
5: **end for**

---

In Algorithm 1, we stop the loop of Line. 3 - 4 if the number of iterations satisfies that:

$$\frac{2(L_f + \gamma L_g)R^2}{(k+1)^2} \leq \epsilon,$$

where $R$ is a constant that satisfies $\|\mathbf{x}_0 - \mathbf{x}^*\| \leq R$.

Combining Theorem 2.5 and [Tseng, 2008, Corollary 2], we establish the following complexity result for problem (P).

**Theorem 3.3.** *Suppose that Assumptions 2.1, 2.2, 3.1 and 3.2 hold and the sequence $\{t_k\}$ in Algorithm 1 satisfies $\frac{1-t_{k+1}}{t_{k+1}^2} \leq \frac{1}{t_k^2}$. Let $\gamma$ be given as in Theorem 2.5. Algorithm 1 generates an $(\epsilon, l_F^{-\beta}\epsilon^\beta)$-optimal solution of problem* (P) *after at most $K$ iterations, where*

$$K = \mathcal{O}\left(\sqrt{\frac{L_{f_1}}{\epsilon}} + \sqrt{\frac{l_F^{\max\{\alpha,\beta\}} L_{g_1}}{\epsilon^{\max\{\alpha,\beta\}}}}\right).$$

Note that Theorem 3.3 encompasses all possible relationships between the magnitudes of $\epsilon_F$ and $\epsilon_G$ in (1), as $\alpha \geq 1$ and $\beta > 0$ are arbitrary. Specially, if $\alpha = 1$ and $\beta \leq \alpha$, the number of iterations is $K = \mathcal{O}\left(\sqrt{(L_{f_1} + l_F L_{g_1})/\epsilon}\right)$. This result matches the lower bound complexity for unconstrained smooth or convex composite optimization [Nemirovsky and Yudin, 1983, Woodworth and Srebro, 2016]. Additionally, if $g_1 \equiv 0$, the number of iterations for obtaining an $(\epsilon, \epsilon^\beta)$-optimal solution of problems (P) is independent of $\gamma$, which can be improved to $K = \mathcal{O}(\sqrt{L_{f_1}/\epsilon})$.

**Remark 3.4.** It is noteworthy that Theorem 1 in a previous paper Samadi et al. [2023] provides the first method that needs $\mathcal{O}(\sqrt{(L_{g_1} + l_F L_{g_1})/\epsilon})$ iterations to achieve an $(\epsilon, \epsilon)$ solution if $\alpha = 1$ and $F$ is smooth. Nevertheless, our methodology diverges in various respects. First, our approach is rooted in the penalization formulation of problem (P$_{\text{Val}}$), while the approach proposed by Samadi et al. [2023] is based on the Tikhonov regularization [Tikhonov and Arsenin, 1977]. Second, we provide a theoretical framework that clearly delineates the relationship between approximate solutions of problems (P) and (P$_\gamma$) for all cases of $\alpha \geq 1$ and $F$ is non-convex, as indicated in Lemmas 2.3, 2.4 and Theorems 2.5, 2.7, 2.8. Therefore, we can first shift our focus from (P) to (P$_\gamma$) based on the penalization framework and then use various methods to solve (P$_\gamma$), not limited to using the APG methods. Besides, the association between approximate solutions of problem (P) and (P$_\gamma$) differs significantly based on whether $\alpha > 1$ or $\alpha = 1$. For the case of $\alpha > 1$, the lower bound comprehensively integrates the accuracy parameter $\epsilon$, which results in a more sophisticated analysis of the convergence result, while Samadi et al. [2023] did not consider the situation when $\alpha > 1$. Third, our method applies to the case that $F$ is composite, while Samadi et al. [2023] requires $F$ to be smooth. Finally, we also propose an adaptive version of our algorithm (see Algorithm 2) that does not require an estimate of $\gamma$.

### 3.1.2 Adaptive version with warm-start mechanism

In practice, the penalty parameter $\gamma$ might be difficult to determine. This motivates us to propose Algorithm 2, which adaptively updates $\gamma$ and invokes PB-APG with dynamic $\gamma$ and solution accuracies.

---

**Algorithm 2** Adaptive PB-APG method (aPB-APG)

---

1: **Input:** $\mathbf{x}_0 \in \mathbb{R}^n$, $\gamma_0 = \gamma_1 > 0$, $L_{f_1}, L_{g_1}, \nu > 1, \eta > 1, \epsilon_0 > 0$.
2: **for** $k \geq 0$ **do**
3: $\quad \phi_k(\mathbf{x}) = f_1(\mathbf{x}) + \gamma_k g_1(\mathbf{x})$
4: $\quad \psi_k(\mathbf{x}) = f_2(\mathbf{x}) + \gamma_k g_2(\mathbf{x})$
5: $\quad$ Invoke $\mathbf{x}_k = \text{PB-APG}(\phi_k, \psi_k, L_{f_1}, L_{g_1}, \mathbf{x}_{k-1}, \epsilon_k)$
6: $\quad \epsilon_{k+1} = \epsilon_k / \eta$
7: $\quad \gamma_{k+1} = \nu \gamma_k$
8: **end for**

---

In Algorithm 2, we adaptively update the penalty parameter $\gamma_k$, and invoke the PB-APG to generate an approximate solution for (P$_\gamma$) with accuracy $\epsilon = \epsilon_k$. Meanwhile, a warm-start mechanism is employed, meaning that the initial point for each subproblem is the output of the preceding subproblem. The convergence result of Algorithm 2 is as follows.

**Theorem 3.5.** *Suppose that Assumptions 2.1, 2.2, 3.1, and 3.2 hold. Also assume that for every outcome of inner loop in Algorithm 2, $\|\mathbf{x}_k - \mathbf{x}_k^*\| \leq R$. Let $\epsilon_0 > 0$ be given.*

- *When $\alpha > 1$, set $\nu > \eta^{\alpha-1}$, and define $N := \lceil \log_{\eta^{1-\alpha}\nu}(\rho L_F^\alpha(\alpha-1)^{\alpha-1}\alpha^{-\alpha}\epsilon_0^{1-\alpha}/\gamma_0) \rceil_+$ and $\gamma_k^* := \rho L_F^\alpha(\alpha-1)^{\alpha-1}\alpha^{-\alpha}\epsilon_0^{1-\alpha}\eta^{k(\alpha-1)}$.*

- *When $\alpha = 1$, set $\nu > 1$, and define $N := \lceil \log_\nu(\rho l_F/\gamma_0) \rceil_+$ and $\gamma_k^* := \rho L_F$.*

*Then, for any $k \geq N$, Algorithm 2 generates an $\left(\frac{\epsilon_0}{\eta^k}, \frac{2\epsilon_0}{\eta^k(\gamma_0\nu^k - \gamma_k^*)}\right)$-optimal solution of problem (P) after at most $K$ iterations, where $K$ satisfies*

$$K = \mathcal{O}\left(\sqrt{\frac{L_{f_1}\eta^k}{\epsilon_0}} + \sqrt{\frac{L_{g_1}\gamma_0(\eta\nu)^k}{\epsilon_0}}\right).$$

Theorem 3.5 shows that for any given initial accuracy $\epsilon_0 > 0$, Algorithm 2 can produce an approximate solution of problem (P) with the desired accuracy.

**Remark 3.6.** From Theorem 3.5, one can obtain an $(\epsilon, \frac{\epsilon}{\gamma_0\nu^k - \gamma_k^*})$-optimal solution of problem (P) within $\mathcal{O}(\sqrt{L_{f_1}/\epsilon} + \sqrt{L_{g_1}/\epsilon^\alpha})$ iterations when $\epsilon/\eta \leq \epsilon_0/\eta^k \leq \epsilon$, which is similar to the complexity results in Theorem 3.3.

### 3.1.3 The upper-level objective is strongly convex

We investigate the convergence outcomes when the smooth part of the upper-level objective exhibits strong convexity.

**Assumption 3.7.** $f_1(\mathbf{x})$ is $\mu$-strongly convex on $\mathrm{dom}(F)$ with $\mu > 0$.

Assumption 3.7 is another widely adopted setting in the existing SBO literature [Beck and Sabach, 2014, Sabach and Shtern, 2017, Amini and Yousefian, 2019, Merchav and Sabach, 2023]. Here, we propose a variant of PB-APG that can provide better complexity results than existing methods. Our main integration is an APG-based algorithm, which has been studied in the existing literature [Nesterov, 2013, Lin and Xiao, 2014, Xu, 2022]. In this paper, we adopt the algorithm proposed in Lin and Xiao [2014] and modify it with a constant step-size for simplicity as in Algorithm 3. Similar to Algorithm 1, we denote Algorithm 3 by $\hat{\mathbf{x}} = \mathrm{PB\text{-}APG\text{-}sc}(\phi_\gamma, \psi_\gamma, \mu, L_{f_1}, L_{g_1}, \mathbf{y}_0, \epsilon)$.

---

**Algorithm 3** PB-APG method for Strong Convexity Case (PB-APG-sc)

---

1: **Input:** $\mu, \gamma, L_\gamma = L_{f_1} + \gamma L_{g_1}, \mathbf{x}_{-1}, \mathbf{y}_0 \in \mathbb{R}^n$.
2: $\tilde{\mathbf{y}} = \mathbf{y}_0 - L_\gamma^{-1}\nabla\phi_\gamma(\mathbf{x}_{-1})$
3: $\tilde{\mathbf{x}} = \mathrm{prox}_{L_\gamma^{-1}\psi_\gamma}(\tilde{\mathbf{y}} - L_\gamma^{-1}\nabla\phi_\gamma(\tilde{\mathbf{y}}))$
4: **Initialization:** Let $\mathbf{x}_{-1} = \mathbf{x}_0 = \tilde{\mathbf{x}}$, $k = 0$
5: **for** $k \geq 0$ **do**
6: $\quad \mathbf{y}_k = \mathbf{x}_k + \frac{\sqrt{L_\gamma} - \sqrt{\mu}}{\sqrt{L_\gamma} + \sqrt{\mu}}(\mathbf{x}_k - \mathbf{x}_{k-1})$
7: $\quad \mathbf{x}_{k+1} = \mathrm{prox}_{L_\gamma^{-1}\psi_\gamma}(\mathbf{y}_k - L_\gamma^{-1}\nabla\phi_\gamma(\mathbf{y}_k))$
8: **end for**

---

The convergence analysis of Algorithm 3 is in the existing literature [Nesterov, 2013, Lin and Xiao, 2014]. Combining [Lin and Xiao, 2014, Theorem 1] and Theorem 2.5, we have the following complexity result.

**Theorem 3.8.** *Suppose that Assumptions 2.1, 2.2, 3.1, 3.2, and 3.7 hold. Algorithm 3 can produce an $(\epsilon, l_F^{-\beta}\epsilon^\beta)$-optimal solution of problem* (P) *after at most $K$ iterations, where $K$ satisfies*

$$K = \mathcal{O}\left(\sqrt{\frac{L_{f_1}}{\mu}}\log\frac{1}{\epsilon} + \sqrt{\frac{l_F^{\max\{\alpha,\beta\}}L_{g_1}}{\epsilon^{\max\{\alpha-1,\beta-1\}}}}\log\frac{1}{\epsilon}\right).$$

Theorem 3.8 improves the complexity results of Theorem 3.3 significantly. Specifically, when $0 < \beta \leq \alpha = 1$, the convergence rate can be improved to be linear, i.e., $K = \mathcal{O}(\sqrt{L_{f_1}/\mu}\log\frac{1}{\epsilon})$.

Additionally, we present an adaptive variant of PB-APG-sc, termed aPB-APG-sc, which adaptively executes $\mathbf{x}_k = \mathrm{PB\text{-}APG\text{-}sc}(\phi_k, \psi_k, \mu, L_{f_1}, L_{g_1}, \mathbf{x}_{k-1}, \epsilon_k)$ and enjoys the similar complexity results of Algorithm 3, as delineated in Algorithm 4 within Appendix D.1.

### 3.2 Both objectives are non-smooth

In this section, we focus on the scenario where both the upper- and lower-level objectives are non-smooth, namely, $f_1 = g_1 \equiv 0$. Additionally, we assume that there is a point $x \in C$ in the lower level problem, where $C$ is either $\mathbb{R}^n$ (the unconstrained case) or a nonempty closed and convex set satisfying $C \subseteq \mathrm{int}\left(\mathrm{dom}(F)\bigcap\mathrm{dom}(G)\right)$.

It is worth noting that in the case where both $F$ and $G$ are non-smooth, the convergence result may not be as favorable as those in the previous scenarios. This is primarily due to the limited availability of information and unfavorable properties concerning $F$ and $G$. In this case, we employ a subgradient method to solve problem $(P_\gamma)$, which has been extensively studied in the existing literature [Shor, 2012, Bubeck et al., 2015, Beck, 2017, Nesterov, 2018]. Specifically, we update

$$\mathbf{x}_{k+1} = \mathrm{Proj}_C(\mathbf{x}_k - \eta_k\xi_k), \tag{4}$$

where $\xi_k \in \partial\Phi_\gamma(\mathbf{x}_k)$ is an subgradient of $\Phi_\gamma(\mathbf{x}_k)$, and $\mathrm{Proj}_C(\mathbf{x})$ is the projection of $\mathbf{x}$ onto $C$.

Let $\mathbf{x}_\gamma^*$ be an optimal solution of problem $(P_\gamma)$ and suppose that there exists a constant $R$ such that $\|\mathbf{x}_0 - \mathbf{x}_\gamma^*\| \leq R$. Motivated by Theorem 8.28 in Beck [2017], we establish the subsequent complexity result for problem (P).

**Theorem 3.9.** *Suppose that Assumption 3.1(3) holds, $f_2$ and $g_2$ are $l_{f_2}$- and $l_{g_2}$-Lipschitz continuous, respectively. Set step-size $\eta_k = \frac{R}{l_\gamma\sqrt{k+1}}$ in* (4). *Then, the subgradient method produces an $(\epsilon, l_{f_2}^{-\beta}\epsilon^\beta)$-optimal*

*solution of problem* (P) *after at most $K$ iterations, where $K$ satisfies*

$$K = \mathcal{O}\left(\frac{l_{f_2}^2}{\epsilon^2} + \frac{l_{f_2}^{\max\{2\alpha, 2\beta\}} l_{g_2}^2}{\epsilon^{\max\{2\alpha, 2\beta\}}}\right).$$

For non-smooth SBO problems, our method has lower complexity compared to existing approaches. Specifically, under a bounded domain assumption, Helou and Simões [2017] simply proposed an $\epsilon$-subgradient method with an asymptotic rate towards the optimal solution set. The a-IRG method in Kaushik and Yousefian [2021] achieved convergence rates of $\mathcal{O}(1/\epsilon^{\frac{1}{0.5-b}})$ and $\mathcal{O}(1/\epsilon^{\frac{1}{b}})$ for the upper- and lower-level objectives, respectively, where $b \in (0, 0.5)$. Setting $b = 0.25$ yields the convergence rates of $\mathcal{O}(1/\epsilon^4)$ for both upper- and lower-level objectives, which indicates that our complexity is more efficient than theirs when $\alpha < 2$ and $\beta \leq \alpha$. Furthermore, the online framework proposed in Shen et al. [2023] performed a complexity of $\mathcal{O}(1/\epsilon^3)$ for both upper- and lower-level objectives. Similarly, our approach prevails over theirs when $\alpha < 1.5$ and $\beta \leq \alpha$.

**Strongly convex upper-level objective.** Based on Theorem 8.31 in Beck [2017], we next explore the improved complexity result for problem (P) when $f_2$ is additionally strongly convex.

**Theorem 3.10.** *Suppose that Assumption 3.1(3) holds, $C \subseteq \text{int}(\text{dom}(F) \bigcap \text{dom}(G))$, $f_2$ is $l_{f_2}$-Lipschitz continuous and $\mu_{f_2}$-strongly convex[3], and $g_2$ is $l_{g_2}$-Lipschitz continuous. Choose step-size $\eta_k = \frac{2}{\mu_{f_2}(k+1)}$ in (4). Then, the subgradient method produces an $(\epsilon, l_{f_2}^{-\beta}\epsilon^\beta)$-optimal solution of problem* (P) *after at most $K$ iterations, where $K$ satisfies*

$$K = \mathcal{O}\left(\frac{l_{f_2}^2}{\mu_{f_2}\epsilon} + \frac{l_{f_2}^{\max\{2\alpha, 2\beta\}} l_{g_2}^2}{\mu_{f_2}\epsilon^{\max\{2\alpha-1, 2\beta-1\}}}\right).$$

To our knowledge, within the context of Theorem 3.10, current findings fail to exploit strong convexity to enhance results. However, our approach capitalizes on distinct structural characteristics that yield superior complexity outcomes relative to Theorem 3.9 in cases where $\alpha < 2$ and $\beta \leq \alpha$.

# 4 Numerical experiments

We apply our Algorithms 1, 2, 3 and 4 to two simple bilevel optimization problems from the motivating examples in Appendix A. The performances of our methods are compared with several existing methods: MNG [Beck and Sabach, 2014], BiG-SAM [Sabach and Shtern, 2017], DBGD [Gong et al., 2021], a-IRG [Kaushik and Yousefian, 2021], CG-BiO [Jiang et al., 2023], Bi-SG [Merchav and Sabach, 2023] and R-APM [Samadi et al., 2023]. For practical efficiency, we use the Greedy FISTA algorithm proposed in Liang et al. [2022] as the APG method in our approach. Detailed settings and additional experimental results are presented in Appendix F.

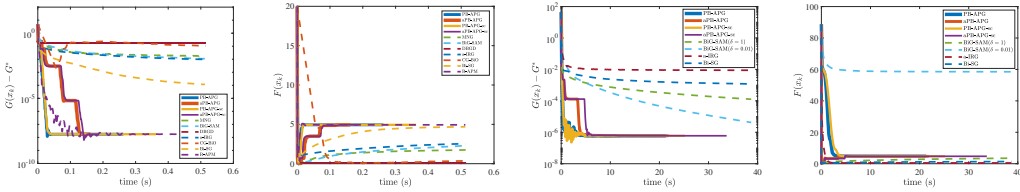

Figure 1: Performances of methods in LRP.   Figure 2: Performances of methods in LSRP.

## 4.1 Logistic regression problem (LRP)

The LRP reads

$$\min_{\mathbf{x} \in \mathbb{R}^n} \frac{1}{2}\|\mathbf{x}\|^2 \quad \text{s.t.} \quad \mathbf{x} \in \arg\min_{\mathbf{z} \in \mathbb{R}^n} \frac{1}{m} \sum_{i=1}^m \log(1 + \exp(-\mathbf{a}_i^{\mathrm{T}}\mathbf{z}b_i)) + I_C(\mathbf{z}), \tag{5}$$

where $I_C(\mathbf{x})$ is the indicator function of the set $C = \{\mathbf{x} \in \mathbb{R}^n : \|\mathbf{x}\|_1 \leq \theta\}$ with $\theta = 10$. Our goal is to find a solution to the lower-level problem with the smallest Euclidean norm. The upper-level objective only

---

[3]In this case, we must have $C$ bounded, as $f_2$ is both strongly convex and Lipschitz continuous.

consists of the smooth part, which is 1-strongly convex and 1-smooth; meanwhile, the lower-level objective is a composite function, where the smooth part is $\frac{1}{4m}\lambda_{\max}(A^{\mathrm{T}}A)$-smooth, and the nonsmooth part is prox-friendly [Duchi et al., 2008].

In this experiment, we compare our methods with MNG, BiG-SAM, DBGD, a-IRG, CG-BiO, and Bi-SG. We plot the values of residuals of the lower-level objective $G(\mathbf{x}_k) - G^*$ and the upper-level objective over time in Figure 1.

As shown in Figure 1, the PB-APG, aPB-APG, PB-APG-sc, and aPB-APG-sc algorithms exhibit significantly faster convergence performance than the other methods for both lower- and upper-level objectives, although R-APM attains similar outcomes, our PB-APG and PB-APG-sc ensure a more rapid decline than it, as shown in the first subfigure of Figure 1. This is because our methods achieve lower optimal gaps and desired function values of the lower- and upper-level objectives with less execution time. This observation confirms the improved complexity results stated in the theorems above. Although the high exactness of our methods for the lower-level problem leads to larger upper-level objectives, Table 3 in Appendix F.1 shows that our methods are much closer to the optimal value. This is reasonable because the other methods exhibit lower accuracy at the lower-level problem, resulting in larger feasible sets compared to the lower-level optimal solution set $X_{\mathrm{opt}}$. In addition, Figure 1 demonstrates that aPB-APG and aPB-APG-sc outperform PB-APG and PB-APG-sc in terms of convergence rate. This improvement can be attributed to the adaptiveness incorporated in Algorithms 2 and 4.

### 4.2 Least squares regression problem (LSRP)

The LSRP has the following form:

$$\min_{\mathbf{x}\in\mathbb{R}^n} \frac{\tau}{2}\|\mathbf{x}\|^2 + \|\mathbf{x}\|_1 \quad \text{s.t.} \quad \mathbf{x} \in \arg\min_{\mathbf{z}\in\mathbb{R}^n} \frac{1}{2m}\|A\mathbf{z}-b\|^2, \tag{6}$$

where $\tau = 0.02$ regulates the trade-off between $\ell_1$ and $\ell_2$ norms. We aim to find a sparse solution for the lower-level problem. The upper-level objective is formulated as a composite function, which consists of a $\tau$-strongly convex and $\tau$-smooth component, along with a proximal-friendly non-smooth component [Beck, 2017]. The lower-level objective is a smooth function with a smoothness parameter of $\frac{1}{m}\lambda_{\max}(A^{\mathrm{T}}A)$.

In this experiment, we compare the performances of our methods with a-IRG, BiG-SAM, and Bi-SG. We plot the values of residuals of lower-level objective $G(\mathbf{x}_k) - G^*$ and the upper-level objective over time in Figure 2.

Figure 2 shows that the proposed PB-APG, aPB-APG, PB-APG-sc, and aPB-APG-sc converge faster than the compared methods for both the lower- and upper-level objectives, as well. For the upper-level objective, our methods achieve larger function values than other methods, except BiG-SAM ($\delta = 0.01$). This is because our methods attain higher accuracy for the lower-level objective than other methods. We have similar observations in Section 4.1. Furthermore, Figure 1 also demonstrates that the adaptive mechanism produces staircase-shaped curves for aPB-APG and aPB-APG-sc, which might prevent undesirable fluctuations in PB-APG and PB-APG-sc.

## 5 Conclusion

This paper proposes a penalization framework that effectively addresses the challenges inherent in simple bilevel optimization problems. By delineating the relationship between approximate solutions of the original problem and its penalized reformulation, we enable the application of specific methods under varying assumptions for the original problem. Under the Hölderian error bound condition, our methods achieve superior complexity results compared to the existing methods. The performance is further improved when the smooth component of the upper-level objective is strongly convex. Additionally, we extend our framework to scenarios involving general nonsmooth objectives. Numerical experiments also validate the effectiveness of our algorithms.

### Acknowledgements

This work is partly supported by the National Key R&D Program of China under grant 2023YFA1009300, National Natural Science Foundation of China under grants 12171100 and the Major Program of NFSC (72394360,72394364).

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

# A Motivating examples

Many machine learning applications involve a primary objective $G$, which usually represents the training loss, and a secondary objective $F$, which can be a regularization term or an auxiliary loss. A common approach for such problems is to optimize $G$ fully and then use $F$ to select the optimal solutions from the ones obtained for $G$. This is called lexicographic optimization [Kissel et al., 2020, Gong et al., 2021]. Two classes of lexicographic optimization problems are the regularized problem, also known as the ill-posed optimization problem [Amini and Yousefian, 2019, Jiang et al., 2023], and the over-parameterized regression [Jiang et al., 2023], where the upper-level objectives are the regularization terms or loss functions, and the lower-level objectives are the loss functions and the constraint terms. We present some examples of these classes of problems as follows.

**Example A.1** (Linear Inverse Problems). Linear inverse problems aim to reconstruct a vector $\mathbf{x} \in \mathbb{R}^n$ from measurements $b \in \mathbb{R}^m$ that satisfy $b = A\mathbf{x} + \rho\varepsilon$, where $A : \mathbb{R}^n \to \mathbb{R}^m$ is a linear mapping, $\varepsilon \in \mathbb{R}^m$ is unknown noise, and $\rho > 0$ is its magnitude. Various optimization techniques can address these problems. We focus on the bilevel formulation, widely adopted in the literature [Beck and Sabach, 2014, Sabach and Shtern, 2017, Dempe et al., 2021, Latafat et al., 2023, Merchav and Sabach, 2023].

The lower-level objective in the bilevel formulation is given by

$$G(\mathbf{x}) = \frac{1}{2m} \|A\mathbf{x} - b\|^2 + I_C(\mathbf{x}), \tag{7}$$

where $I_C(\mathbf{x})$ is the indicator function of a set $C$ that satifies $I_C(\mathbf{x}) = 0$ if $\mathbf{x} \in C$, and $I_C(\mathbf{x}) = +\infty$ if $\mathbf{x} \notin C$. The set $C$ is a closed, convex set that can be chosen as $C = \mathbb{R}^n$, $C = \{\mathbf{x} \in \mathbb{R}^n : \mathbf{x} \geq 0\}$, or $C = \{\mathbf{x} \in \mathbb{R}^n : \|\mathbf{x}\|_1 \leq \theta\}$ for some $\theta > 0$.

This problem may have multiple minimizer solutions. Hence, a reasonable option is to consider the minimal norm solution problem, i.e., find the optimal solution with the smallest Euclidean norm [Beck and Sabach, 2014, Sabach and Shtern, 2017, Latafat et al., 2023]:

$$F(\mathbf{x}) = \frac{1}{2} \|\mathbf{x}\|^2 .$$

We need to solve the simple bilevel optimization problem:

$$\min_{\mathbf{x} \in \mathbb{R}^n} \frac{1}{2} \|\mathbf{x}\|^2 \quad \text{s.t.} \quad \mathbf{x} \in \arg\min_{\mathbf{z} \in \mathbb{R}^n} \frac{1}{2m} \|A\mathbf{z} - b\|^2 + I_C(\mathbf{z}).$$

**Example A.2** (Sparse Solution of Linear Inverse Problems). Consider the same setting as in Example A.1, but with the additional goal of finding a sparse solution among all the minimizers of the linear inverse problem (7). This can simplify the model and improve computational efficiency. To achieve sparsity, we can use any function that encourages it. One such function is the well-known elastic net regularization [Friedlander and Tseng, 2008, Amini and Yousefian, 2019, Merchav and Sabach, 2023], which is defined as

$$F(\mathbf{x}) = \|\mathbf{x}\|_1 + \frac{\tau}{2} \|\mathbf{x}\|^2 ,$$

where $\tau > 0$ regulates the trade-off between $\ell_1$ and $\ell_2$ norms.

This example corresponds to our second experiment in Section 4.2.

**Example A.3** (Logistic Regression Problem). The logistic regression problem aims to map the feature vectors $\mathbf{a}_i$ to the target labels $b_i$. A standard machine learning technique for this problem is to minimize the logistic loss function over the given dataset [Amini and Yousefian, 2019, Gong et al., 2021, Jiang et al., 2023, Latafat et al., 2023, Merchav and Sabach, 2023]. We assume that the dataset consists of a feature matrix $A \in \mathbb{R}^{m \times n}$ and a label vector $b \in \mathbb{R}^m$, with $b_i \in \{-1, 1\}$ for each $i$. The logistic loss function is defined as

$$g_1(\mathbf{x}) = \frac{1}{m} \sum_{i=1}^{m} \log(1 + \exp(-\mathbf{a}_i^{\mathrm{T}} \mathbf{x} b_i)). \tag{8}$$

Over-fitting is a common issue when the number of features is large compared to the number of instances $m$. A possible approach is to regularize the logistic objective function with a specific function or a constraint [Jiang et al., 2023, Merchav and Sabach, 2023]. For instance, we can use $g_2(\mathbf{x}) = I_C(\mathbf{x})$, where $I_C(\mathbf{x})$ is the indicator of the set $C = \{\mathbf{x} \in \mathbb{R}^n : \|\mathbf{x}\|_1 \leq \theta\}$, as in Example A.1.

This problem may also have multiple optimal solutions. Hence, a natural extension is to consider the minimal norm solution problem [Gong et al., 2021, Jiang et al., 2023, Latafat et al., 2023], as in Example A.1. This requires solving the following problem:

$$\min_{\mathbf{x} \in \mathbb{R}^n} \frac{1}{2} \|\mathbf{x}\|^2 \quad \text{s.t.} \quad \mathbf{x} \in \arg\min_{\mathbf{z} \in \mathbb{R}^n} \frac{1}{m} \sum_{i=1}^{m} \log(1 + \exp(-\mathbf{a}_i^{\mathrm{T}} \mathbf{z} b_i)) + I_C(\mathbf{z}).$$

When choosing $C = \{\mathbf{x} \in \mathbb{R}^n : \|\mathbf{x}\|_1 \leq \theta\}$ for some $\theta > 0$, it corresponds to our first experiment in Section 4.1.

**Example A.4** (Over-parameterized Regression Problem). The linear regression problem aims to find a parameter vector $\mathbf{x} \in \mathbb{R}^n$ that minimizes the training loss $\ell_{\mathrm{tr}}(\mathbf{x})$ over the training dataset $\mathcal{D}_{\mathrm{tr}}$. Without explicit regularization, the over-parameterized regression problem has multiple minima. However, these minima may have different generalization performance. Therefore, we introduce a secondary objective, such as the validation loss over a validation set $\mathcal{D}_{\mathrm{val}}$, to select one of the global minima of the training loss. This results in the following bilevel problem:

$$\min_{\mathbf{x} \in \mathbb{R}^n} F(\mathbf{x}) := \ell_{\mathrm{val}}(\mathbf{x}) \quad \text{s.t.} \quad \mathbf{x} \in \arg\min_{\mathbf{z} \in \mathbb{R}^n} G(\mathbf{z}) := \ell_{\mathrm{tr}}(\mathbf{z}). \tag{9}$$

For instance, we can consider the sparse linear regression problem, where the lower-level objective consists of the training error and a regularization term, namely, $G(\mathbf{x}) = \frac{1}{2}\|A_{\mathrm{tr}}\mathbf{x} - b_{\mathrm{tr}}\|^2 + I_C(\mathbf{x})$. Here, $I_C(\mathbf{x})$ denotes the indicator of a convex set, as in Example A.2. The upper-level objective is the validation error, i.e., $F(\mathbf{x}) = \frac{1}{2}\|A_{\mathrm{val}}\mathbf{x} - b_{\mathrm{val}}\|^2$. The linear regression problem is over-parameterized when the number of features $n$ is larger than the number of data instances in the training set.

# B  Comparison between simple bilevel optimization methods

Table 1: Summary of simple bilevel optimization algorithms. The abbreviations "SC," "C,", "diff", "comp", "WS" and "C3" represent "strongly convex," "convex,", "differentiable", "composite", "weak sharpness" and "Convex objective with Convex Compact constraints," respectively. The abbreviation $\alpha$-HEB refers to Hölderian error bound with exponent parameter $\alpha$. We only include the gradients Lipschitz constant in the complexity result when its relation to the complexity is clear; otherwise, we omit it. Notation $l_F$ is the upper bound of subdifferentials of $F$, $L_{f_1}$ and $L_{g_1}$ are the Lipschitz constants of $\nabla f_1$ and $\nabla g_1$, respectively.

| Methods | Upper-level Objective $F$ | Lower-level Objective $G$ | $(\epsilon_F, \epsilon_G)$-optimal Solution | Convergence Upper-level | Convergence Lower-level |
|---|---|---|---|---|---|
| MNG [Beck and Sabach, 2014] | SC, diff | C, smooth | $(/, \epsilon_G)$ | Asymptotic | $\mathcal{O}\left(L_{g_1}^2/\epsilon_G^2\right)$ |
| BiG-SAM [Sabach and Shtern, 2017] | SC, smooth | C, comp | $(/, \epsilon_G)$ | Asymptotic | $\mathcal{O}\left(L_{g_1}/\epsilon_G\right)$ |
| IR-IG [Amini and Yousefian, 2019] | SC | C3, Finite sum | $(/, \epsilon_G)$ | Asymptotic | $\mathcal{O}\left(1/\epsilon_G^{\frac{1}{0.5-\varepsilon}}\right), \varepsilon \in (0, 0.5)$ |
| IR-CG [Giang-Tran et al., 2023] | C, smooth | C3, smooth | $(\epsilon_F, \epsilon_G)$ | | $\mathcal{O}\left(\max\{1/\epsilon_F^{\frac{1}{1-p}}, 1/\epsilon_G^{\frac{1}{p}}\}\right) p \in (0, 1)$ |
| Tseng's method [Malitsky, 2017] | C, comp | C, comp | $(/, \epsilon_G)$ | Asymptotic | $\mathcal{O}\left(1/\epsilon_G\right)$ |
| ITALEX [Doron and Shtern, 2023] | C, comp | C, comp | $(\epsilon, \epsilon^2)$ | | $\mathcal{O}\left(1/\epsilon^2\right)$ |
| a-IRG [Kaushik and Yousefian, 2021] | C, Lip | C, Lip | $(\epsilon_F, \epsilon_G)$ | | $\mathcal{O}\left(\max\{1/\epsilon_F^{\frac{1}{0.5-b}}, 1/\epsilon_G^{\frac{1}{b}}\}\right), b \in (0, 0.5)$ |
| CG-BiO [Jiang et al., 2023] | C, smooth | C3, smooth | $(\epsilon_F, \epsilon_G)$ | | $\mathcal{O}(\max\{L_{f_1}/\epsilon_F, L_{g_1}/\epsilon_G\})$ |
| Bi-SG [Merchav and Sabach, 2023] | C, quasi-Lip/comp | C, comp | $(\epsilon_F, \epsilon_G)$ | | $\mathcal{O}\left(\max\{1/\epsilon_F^{\frac{1}{1-a}}, 1/\epsilon_G^{\frac{1}{a}}\}\right), a \in (0.5, 1)$ |
| | $\mu$-SC, comp | C, comp | $(\epsilon_F, \epsilon_G)$ | | $\mathcal{O}\left(\max\{\left(\frac{\log 1/\epsilon_F}{\mu}\right)^{\frac{1}{1-a}}, 1/\epsilon_G^{\frac{1}{a}}\}\right), a \in (0.5, 1)$ |
| R-APM [Samadi et al., 2023] | C, smooth | C, comp, WS | $(\epsilon, \epsilon)$ | | $\mathcal{O}\left(\sqrt{1/\epsilon}\right)$ |
| Online Framework [Shen et al., 2023] | C, Lip | C3, Lip | $(\epsilon_F, \epsilon_G)$ | | $\mathcal{O}\left(\max\{1/\epsilon_F^3, 1/\epsilon_G^3\}\right)$ |
| **Our method** | C, comp | C, comp, $\alpha$-HEB | $(\epsilon, l_F^{-\beta}\epsilon^\beta)$ | | $\mathcal{O}\left(\sqrt{\frac{L_{f_1}}{\epsilon}} + \sqrt{\frac{l_F^{\max\{\alpha,\beta\}}L_{g_1}}{\epsilon^{\max\{\alpha,\beta\}}}}\right), \alpha \geq 1, \beta > 0$ |
| | $\mu$-SC, comp | C, comp, $\alpha$-HEB | $(\epsilon, l_F^{-\beta}\epsilon^\beta)$ | | $\mathcal{O}\left(\sqrt{\frac{L_{f_1}}{\mu}}\log\frac{1}{\epsilon} + \sqrt{\frac{l_F^{\max\{\alpha,\beta\}}L_{g_1}}{\epsilon^{\max\{\alpha-1,\beta-1\}}}}\log\frac{1}{\epsilon}\right), \alpha \geq 1, \beta > 0$ |
| | nonsmooth, Lip | nonsmooth, Lip, $\alpha$-HEB | $(\epsilon, l_F^{-\beta}\epsilon^\beta)$ | | $\mathcal{O}\left(\frac{l_{f_2}^2}{\epsilon^2} + \frac{l_{f_2}^{\max\{2\alpha,2\beta\}}l_{g_2}^2}{\epsilon^{\max\{2\alpha,2\beta\}}}\right), \alpha \geq 1, \beta > 0$ |

# C  Examples of functions satisfying the Hölderian error bound

We present several examples of functions that satisfy the Hölderian error bound Assumption 2.2 and their corresponding exponent parameter $\alpha$ in Table 2. We also provide some clarifications for Table 2 below. The abbreviations "$Q \in \mathbb{S}^n$" and "$Q \succ 0$" stand for "$Q$ is a symmetric matrix of order $n$ and a positive definite matrix, respectively. We refer the reader to Pang [1997], Bolte et al. [2017], Zhou and So [2017], Jiang and Li [2022], Doron and Shtern [2023] and the references therein for more examples of functions that satisfy Hölderian error bound Assumption 2.2. Furthermore, it is noteworthy that numerous applications in neural networks, such as deep neural networks (DNNs), also comply with this assumption, as discussed in Bolte et al. [2017], Zeng et al. [2019].

---

[4]According to Table 2 of Doron and Shtern [2023], the parameter $\alpha$ can take values of either 1 or 2. Particularly, when $\alpha = 1$, we have $\rho = 1$; when $\alpha = 2$, we have $\rho = 2/\tau$.

Table 2: Summary of some functions satisfying Hölderian error bound with corresponding exponents.

| $G(\mathbf{x})$ | Remarks | Name | $\alpha$ |
|---|---|---|---|
| $\max_{i\in[m]}\{\langle\mathbf{a}_i,\mathbf{x}\rangle - b_i\}$ | $\mathbf{a}_i\in\mathbb{R}^n, i\in[m], b\in\mathbb{R}^m$ | piece-wise maximum | 1 |
| $\|\mathbf{x}-\mathbf{x}_0\|_Q = \sqrt{(\mathbf{x}-\mathbf{x}_0)^{\mathrm{T}}Q(\mathbf{x}-\mathbf{x}_0)}$ | $Q\in\mathbb{S}^n, Q\succ 0, \mathbf{x}_0\in\mathbb{R}^n$ | $Q$-norm | 1 |
| $\|\mathbf{x}-\mathbf{x}_0\|_p$ | $\mathbf{x}_0\in\mathbb{R}^n, p\geq 1$ | $\ell_p$-norm | 1 |
| $\|x\|_1 + \frac{\tau}{2}\|x\|^2$ | $\tau > 0$ | Elastic net | 1 or $2^4$ |
| $\|A\mathbf{x}-b\|^2$ | $A\in\mathbb{R}^{m\times n}, b\in\mathbb{R}^m$ | Least squares | 2 |
| $\frac{1}{m}\sum_{i=1}^m \log(1+\exp(-\mathbf{a}_i^{\mathrm{T}}\mathbf{x}b_i))$ | $\mathbf{a}_i\in\mathbb{R}^n, i\in[m], b\in\mathbb{R}^m, A\in\mathbb{R}^{m\times n}$ | Logistic loss | 2 |
| $\eta(\mathbf{x}) + \frac{\sigma}{2}\|\mathbf{x}\|^2$ | $\eta$ convex, $\sigma > 0$ | Strongly-convex | 2 |

# D    Supplementary results

## D.1    Adaptive version of PB-APG method with strong convexity assumption

---
**Algorithm 4** Adaptive PB-APG-sc method (aPB-APG-sc)

---
1: **Input:** $\mathbf{x}_{-1} = \mathbf{x}_0 \in \mathbb{R}^n$, $\gamma_0 = \gamma_1 > 0$, $L_{f_1}, L_{g_1}, \nu > 1, \eta > 1, \epsilon_0 > 0$.
2: **for** $k \geq 0$ **do**
3:      $\phi_k(\mathbf{x}) = f_1(\mathbf{x}) + \gamma_k g_1(\mathbf{x})$
4:      $\psi_k(\mathbf{x}) = f_2(\mathbf{x}) + \gamma_k g_2(\mathbf{x})$
5:      Invoke $\mathbf{x}_k = \text{PB-APG-sc}(\phi_k, \psi_k, \mu, L_{f_1}, L_{g_1}, \mathbf{x}_{k-1}, \epsilon_k)$
6:      $\epsilon_{k+1} = \frac{1}{\eta}\epsilon_k$
7:      $\gamma_{k+1} = \nu\gamma_k$
8: **end for**

---

Similar to Algorithm 2, we have the following convergence results of Algorithm 4.

**Theorem D.1.** *Suppose that Assumptions 2.1, 2.2, 3.1, 3.2, and 3.7 hold. Let $\epsilon_0 > 0$ be given.*

- *When $\alpha > 1$, set $\nu > \eta^{\alpha-1}$, $N = \lceil \log_{\eta^{1-\alpha}\nu}(\rho L_F^\alpha(\alpha-1)^{\alpha-1}\alpha^{-\alpha}\epsilon_0^{1-\alpha}/\gamma_0)\rceil_+$ and $\gamma_k^* = \rho L_F^\alpha(\alpha-1)^{\alpha-1}\alpha^{-\alpha}\epsilon_0^{1-\alpha}\eta^{k(\alpha-1)}$;*

- *When $\alpha = 1$, set $\nu > 1$, $N = \lceil \log_\nu(\rho l_F/\gamma_0)\rceil_+$ and $\gamma_k^* = \rho L_F$.*

*Then, for any $k \geq N$, Algorithm 2 generates an $(\frac{\epsilon_0}{\eta^k}, \frac{2\epsilon_0}{\eta^k(\gamma_0\nu^k - \gamma_k^*)})$-optimal solution of problem* (P) *after at most $K$ iterations, where $K$ satisfies*

$$K = \mathcal{O}\left(\sqrt{\frac{L_{f_1}}{\mu}}\log\frac{\eta^k}{\epsilon_0} + \sqrt{\frac{\nu^k l_F^{\max\{\alpha,\beta\}}L_{g_1}}{\epsilon^{\max\{\alpha-1,\beta-1\}}}}\log\frac{\eta^k}{\epsilon_0}\right).$$

The proof is similar to the proof of Theorem 3.5 in Appendix E.8. So we omit it here.

# E    Proofs of main results

In this section, we propose the proofs of our main convergence results in this paper.

## E.1    Proof of Lemma 2.3

*Proof.* Since $X_{\text{opt}}$ is closed and convex [Beck and Sabach, 2014], the projection of any $\mathbf{x} \in \mathbb{R}^n$ onto $X_{\text{opt}}$, denoted as $\bar{\mathbf{x}}$, exists and is unique. Furthermore, it holds that $\text{dist}(\mathbf{x}, X_{\text{opt}}) = \|\mathbf{x} - \bar{\mathbf{x}}\|$.

Then, by Assumption 2.1, we have

$$F(\mathbf{x}) - F(\bar{\mathbf{x}}) \geq -\xi^\top(\mathbf{x}-\bar{\mathbf{x}}) \geq -\|\xi\|\|\mathbf{x}-\bar{\mathbf{x}}\| \geq -l_F\|\mathbf{x}-\bar{\mathbf{x}}\|, \ \forall\xi\in\partial F(\bar{\mathbf{x}}). \tag{10}$$

Choosing $\gamma^* = \rho l_F^\alpha (\alpha - 1)^{\alpha-1} \alpha^{-\alpha} \epsilon^{1-\alpha}$, it follows that

$$
\begin{aligned}
F(\mathbf{x}) - F(\bar{\mathbf{x}}) + \gamma^* p(\mathbf{x}) &\overset{(10)}{\geq} -l_F \|\mathbf{x} - \bar{\mathbf{x}}\| + \gamma^* p(\mathbf{x}) \\
&\overset{(a)}{\geq} -l_F \|\mathbf{x} - \bar{\mathbf{x}}\| + \frac{\gamma^*}{\rho} \|\mathbf{x} - \bar{\mathbf{x}}\|^\alpha \\
&\geq \min_{\mathbf{z} \geq 0} -l_F \mathbf{z} + \frac{\gamma^*}{\rho} \mathbf{z}^\alpha \\
&\overset{(b)}{=} -\epsilon,
\end{aligned}
\tag{11}
$$

where $(a)$ follows from the Hölderian error bound assumption of $p(\mathbf{x})$, and $(b)$ is from the fact that $\mathbf{y} = -l_F \mathbf{z} + \frac{\gamma^*}{\rho} \mathbf{z}^\alpha$ attains its minimum at $\mathbf{z}^* = \left( \frac{\rho l_F}{\alpha \gamma^*} \right)^{\frac{1}{\alpha-1}}$.

Since $\bar{\mathbf{x}} \in X_{\text{opt}}$ is feasible for problem (P), we have $F(\bar{\mathbf{x}}) \geq F^*$. This along with (11) indicates

$$
F(\mathbf{x}) + \gamma p(\mathbf{x}) - F^* \geq F(\mathbf{x}) + \gamma^* p(\mathbf{x}) - F(\bar{\mathbf{x}}) \geq -\epsilon, \quad \forall \mathbf{x} \in \mathbb{R}^d \text{ and } \gamma \geq \gamma^*.
\tag{12}
$$

Let $\mathbf{x}^*$ be an optimal solution of (P) so that $F(\mathbf{x}^*) = F^*$. In addition, since $\mathbf{x}^* \in X_{\text{opt}}$, we have $p(\mathbf{x}^*) = 0$. Combine these results with (12), we have

$$
F(\mathbf{x}^*) + \gamma p(\mathbf{x}^*) = F^* \overset{(12)}{\leq} F(\mathbf{x}) + \gamma p(\mathbf{x}) + \epsilon, \quad \forall \mathbf{x} \in \mathbb{R}^d \text{ and } \gamma \geq \gamma^*.
\tag{13}
$$

This demonstrates that an optimal solution of (P) is an $\epsilon$-optimal solution for $(P_\gamma)$. $\qquad\square$

## E.2 Proof of Lemma 2.4

*Proof.* The proof is motivated by Theorem 1 in Luo et al. [1996]. Denote $\mathbf{x}^*, \mathbf{x}_\gamma^*$ as optimal solutions of problem (P) and $(P_\gamma)$, respectively.

For any $\mathbf{x} \in \mathbb{R}^n$, let $\bar{\mathbf{x}}$ be the projection of $\mathbf{x}$ onto $X_{\text{opt}}$. Then $\bar{\mathbf{x}}$ is a feasible solution of (P) and $F(\bar{\mathbf{x}}) \geq F(\mathbf{x}^*)$ holds. Then we have

$$
\begin{aligned}
F(\mathbf{x}) + \gamma p(\mathbf{x}) &= F(\bar{\mathbf{x}}) + F(\mathbf{x}) - F(\bar{\mathbf{x}}) + \gamma p(\mathbf{x}) \\
&\geq F(\mathbf{x}^*) + F(\mathbf{x}) - F(\bar{\mathbf{x}}) + \gamma p(\mathbf{x}) \\
&\overset{(a)}{\geq} F(\mathbf{x}^*) - l_F \|\mathbf{x} - \bar{\mathbf{x}}\| + \frac{\gamma}{\rho} \|\mathbf{x} - \bar{\mathbf{x}}\| \\
&= F(\mathbf{x}^*) + (\frac{\gamma}{\rho} - l_F) \|\mathbf{x} - \bar{\mathbf{x}}\| \\
&\overset{(b)}{\geq} F(\mathbf{x}^*) = F(\mathbf{x}^*) + \gamma p(\mathbf{x}^*),
\end{aligned}
\tag{14}
$$

where $(a)$ follows from (10) and the Hölderian error bound assumption of $p(\mathbf{x})$, and $(b)$ follows from $\gamma \geq \rho l_F$. Therefore, we conclude that $\mathbf{x}^*$ is an optimal solution of $(P_\gamma)$.

For the converse part, let $\bar{\mathbf{x}}_\gamma^*$ be the projection of $\mathbf{x}_\gamma^*$ onto $X_{\text{opt}}$. Then $\bar{\mathbf{x}}_\gamma^*$ is a feasible solution of (P). Therefore, it holds that $F(\bar{\mathbf{x}}_\gamma^*) \geq F(\mathbf{x}^*)$. Similarly, we have

$$
\begin{aligned}
F(\mathbf{x}^*) &= F(\mathbf{x}^*) + \gamma p(\mathbf{x}^*) \\
&\geq F(\mathbf{x}_\gamma^*) + \gamma p(\mathbf{x}_\gamma^*) \\
&= F(\mathbf{x}_\gamma^*) - F(\mathbf{x}^*) + F(\mathbf{x}^*) + \gamma p(\mathbf{x}_\gamma^*) \\
&\geq F(\mathbf{x}^*) + F(\mathbf{x}_\gamma^*) - F(\bar{\mathbf{x}}_\gamma^*) + \gamma p(\mathbf{x}_\gamma^*) \\
&\overset{(c)}{\geq} F(\mathbf{x}^*) - l_F \|\mathbf{x}_\gamma^* - \bar{\mathbf{x}}_\gamma^*\| + \frac{\gamma}{\rho} \|\mathbf{x}_\gamma^* - \bar{\mathbf{x}}_\gamma^*\| \\
&\geq F(\mathbf{x}^*) + (\frac{\gamma}{\rho} - l_F) \|\mathbf{x}_\gamma^* - \bar{\mathbf{x}}_\gamma^*\| \\
&\geq F(\mathbf{x}^*),
\end{aligned}
\tag{15}
$$

where the inequality $(c)$ follows from (10) and the Hölderian error bound assumption of $p(\mathbf{x})$.

Therefore, all inequalities in (15) become equalities. We deduce that $\|\mathbf{x}_\gamma^* - \bar{\mathbf{x}}_\gamma^*\| = 0$ if $\gamma > \rho l_F$, implying that $\mathbf{x}_\gamma^*$ is in $X_{\text{opt}}$, i.e., $p(\mathbf{x}_\gamma^*) = 0$. Furthermore, as the first inequality of (15) becomes an equality, we obtain

$$
F(\mathbf{x}^*) = F(\mathbf{x}_\gamma^*) + \gamma p(\mathbf{x}_\gamma^*) = F(\mathbf{x}_\gamma^*).
$$

Therefore, $\mathbf{x}_\gamma^*$ is also an optimal solution of (P). $\qquad\square$

### E.3 Proof of Theorem 2.5

*Proof.* Denote $\mathbf{x}^*$, $\mathbf{x}_\gamma^*$ as optimal solutions of problem (P) and ($P_\gamma$), respectively.

- **Case of $\alpha > 1$.** Since $\tilde{\mathbf{x}}_\gamma^*$ is an $\epsilon$-optimal solution of ($P_\gamma$), we have
$$F(\tilde{\mathbf{x}}_\gamma^*) + \gamma p(\tilde{\mathbf{x}}_\gamma^*) \le F(\mathbf{x}) + \gamma p(\mathbf{x}) + \epsilon, \quad \forall \mathbf{x} \in \mathbb{R}^n. \tag{16}$$
Note that the arguments in the proof of Lemma 2.3 still hold. Substituting $\mathbf{x} = \mathbf{x}^*$ into (16) and utilizing $p(\mathbf{x}^*) = 0$, we have
$$F(\tilde{\mathbf{x}}_\gamma^*) + \gamma p(\tilde{\mathbf{x}}_\gamma^*) \le F(\mathbf{x}^*) + \epsilon = F(\mathbf{x}^*) + \gamma^* p(\mathbf{x}^*) + \epsilon \le F(\tilde{\mathbf{x}}_\gamma^*) + \gamma^* p(\tilde{\mathbf{x}}_\gamma^*) + 2\epsilon,$$
where the last inequality follows from setting $\mathbf{x} = \tilde{\mathbf{x}}_\gamma^*$ in (13). Then, it holds that
$$p(\tilde{\mathbf{x}}_\gamma^*) \le \frac{2\epsilon}{\gamma - \gamma^*} = \frac{2\epsilon}{2 l_F^\beta \epsilon^{1-\beta}} = l_F^{-\beta} \epsilon^\beta. \tag{17}$$
By setting $\mathbf{x} = \mathbf{x}^*$ in (16), we have
$$F(\tilde{\mathbf{x}}_\gamma^*) - F(\mathbf{x}^*) \le \gamma(p(\mathbf{x}^*) - p(\tilde{\mathbf{x}}_\gamma^*)) + \epsilon.$$
Using the fact that $p(\mathbf{x}^*) = 0 \le p(\tilde{\mathbf{x}}_\gamma^*)$, we have
$$F(\tilde{\mathbf{x}}_\gamma^*) - F(\mathbf{x}^*) \le \epsilon. \tag{18}$$
Combing (18) with (17), we conclude that $\tilde{\mathbf{x}}_\gamma^*$ is an $(\epsilon, l_F^{-\beta} \epsilon^\beta)$-optimal solution of (P).

- **Case of $\alpha = 1$.** Since $\tilde{\mathbf{x}}_\gamma^*$ is an $\epsilon$-optimal solution of ($P_\gamma$), we have
$$F(\tilde{\mathbf{x}}_\gamma^*) + \gamma p(\tilde{\mathbf{x}}_\gamma^*) \le F(\mathbf{x}_\gamma^*) + \gamma p(\mathbf{x}_\gamma^*) + \epsilon. \tag{19}$$

On the one hand, as $\gamma = \gamma^* + l_F^\beta \epsilon^{1-\beta} > \gamma^*$, by Lemma 2.4, $\mathbf{x}_\gamma^*$ is an optimal solution of (P). On the other hand, since $\gamma \ge \gamma^*$, according to Lemma 2.4, $\mathbf{x}^*$ is also an optimal solution of ($P_\gamma$). Therefore, $p(\mathbf{x}^*) = 0$ and $p(\mathbf{x}_\gamma^*) = 0$, it holds that
$$\begin{aligned}
F(\mathbf{x}^*) &\le F(\tilde{\mathbf{x}}_\gamma^*) + \gamma p(\tilde{\mathbf{x}}_\gamma^*) \\
&\overset{(19)}{\le} F(\mathbf{x}_\gamma^*) + \gamma p(\mathbf{x}_\gamma^*) + \epsilon \\
&= F(\mathbf{x}_\gamma^*) + \gamma^* p(\mathbf{x}_\gamma^*) + \epsilon \\
&= F(\mathbf{x}^*) + \gamma^* p(\mathbf{x}^*) + \epsilon \\
&\le F(\tilde{\mathbf{x}}_\gamma^*) + \gamma^* p(\tilde{\mathbf{x}}_\gamma^*) + \epsilon,
\end{aligned} \tag{20}$$
where the first inequality follows from the fact that $\mathbf{x}^*$ is an optimal solution of ($P_\gamma$), and the last inequality follows from the optimality of $\mathbf{x}^*$ to ($P_\gamma$) when $\gamma \ge \gamma^*$.

The second inequality of (20) and $p(\tilde{\mathbf{x}}_\gamma^*) \ge 0$ imply that
$$F(\tilde{\mathbf{x}}_\gamma^*) \le F(\mathbf{x}_\gamma^*) + \gamma p(\mathbf{x}_\gamma^*) + \epsilon = F(\mathbf{x}^*) + \gamma p(\mathbf{x}^*) + \epsilon \le F(\mathbf{x}^*) + \epsilon.$$
That is, it holds that
$$F(\tilde{\mathbf{x}}_\gamma^*) \le F(\mathbf{x}^*) + \epsilon. \tag{21}$$

In addition, from (20), we have $F(\tilde{\mathbf{x}}_\gamma^*) + \gamma p(\tilde{\mathbf{x}}_\gamma^*) \le F(\tilde{\mathbf{x}}_\gamma^*) + \gamma^* p(\tilde{\mathbf{x}}_\gamma^*) + \epsilon$, which implies that
$$p(\tilde{\mathbf{x}}_\gamma^*) \le \frac{\epsilon}{\gamma - \gamma^*} = \frac{\epsilon}{l_F^\beta \epsilon^{1-\beta}} = l_F^{-\beta} \epsilon^\beta. \tag{22}$$

This result along with (21) demonstrate that $\tilde{\mathbf{x}}_\gamma^*$ is an $(\epsilon, l_F^{-\beta} \epsilon^\beta)$-optimal solution of (P).

$\square$

### E.4 Proof of Theorem 2.6

*Proof.* Let $\hat{\mathbf{x}}_\gamma^*$ be the projection of $\tilde{\mathbf{x}}_\gamma^*$ on $X_{\text{opt}}$, we have $\|\tilde{\mathbf{x}}_\gamma^* - \hat{\mathbf{x}}_\gamma^*\| = \text{dist}(\tilde{\mathbf{x}}_\gamma^*, X_{\text{opt}})$.

By Assumption 2.2, the following inequality holds,
$$\|\tilde{\mathbf{x}}_\gamma^* - \hat{\mathbf{x}}_\gamma^*\|^\alpha \le \rho p(\tilde{\mathbf{x}}_\gamma^*) \overset{(a)}{\le} \rho l_F^{-\beta} \epsilon^\beta \implies \|\tilde{\mathbf{x}}_\gamma^* - \hat{\mathbf{x}}_\gamma^*\| \le \left(\rho l_F^{-\beta} \epsilon^\beta\right)^{\frac{1}{\alpha}}, \tag{23}$$
where $(a)$ follows from (17) when $\alpha > 1$ or from (22) when $\alpha = 1$.

By Assumption 2.1, we have
$$F(\tilde{\mathbf{x}}_\gamma^*) - F^* \ge F(\tilde{\mathbf{x}}_\gamma^*) - F(\hat{\mathbf{x}}_\gamma^*) \overset{(10)}{\ge} -l_F \|\tilde{\mathbf{x}}_\gamma^* - \hat{\mathbf{x}}_\gamma^*\| \ge -l_F \left(\rho l_F^{-\beta} \epsilon^\beta\right)^{\frac{1}{\alpha}},$$
where the first inequality follows from $F(\hat{\mathbf{x}}_\gamma^*) \ge F^*$ and $\hat{\mathbf{x}}_\gamma^* \in X_{\text{opt}}$.

$\square$

## E.5   Proof of Theorem 2.7

*Proof.* For any $\mathbf{x} \in \mathrm{dom}(F)$, let $\bar{\mathbf{x}}$ be the projection of $\mathbf{x}$ onto $X_{\mathrm{opt}}$, where the existence and uniqueness of $\bar{\mathbf{x}}$ follows from that $X_{\mathrm{opt}}$ is closed and convex. Since $F$ is $l$-Lipschitz continuous, similar to (10), we have

$$F(\mathbf{x}) - F(\bar{\mathbf{x}}) \geq -l\|\mathbf{x} - \bar{\mathbf{x}}\|, \ \forall \xi \in \partial F(\bar{\mathbf{x}}). \tag{24}$$

Therefore, all the requirements of (10) in equations (11), (14) and (15) can be replaced by (24). This implies that Lemmas 2.3 and 2.4 also hold for the global solutions of problems (P) and (P$_\gamma$) when $F$ is non-convex. Then, the final result follows a similar pattern to Theorem 2.5. Here we omit it.   □

## E.6   Proof of Theorem 2.8

*Proof.* Let $\bar{\mathbf{x}}^*_\gamma$ be the projection of $\mathbf{x}^*_\gamma$ onto $X_{\mathrm{opt}}$ and $\hat{\mathbf{x}}^*_\gamma = c\mathbf{x}^*_\gamma + (1-c)\bar{\mathbf{x}}^*_\gamma$ with $c = \min\{1, 1 - \frac{r}{\|\mathbf{x}^*_\gamma - \bar{\mathbf{x}}^*_\gamma\|}\}$, which implies that $\hat{\mathbf{x}}^*_\gamma \in \mathcal{B}(\mathbf{x}^*_\gamma, r)$. Then, we have

$$F(\mathbf{x}^*_\gamma) + \gamma p(\mathbf{x}^*_\gamma) \leq F(\hat{\mathbf{x}}^*_\gamma) + \gamma p(\hat{\mathbf{x}}^*_\gamma) \overset{(i)}{\leq} F(\hat{\mathbf{x}}^*_\gamma) + \gamma(cp(\mathbf{x}^*_\gamma) + (1-c)p(\bar{\mathbf{x}}^*_\gamma)) = F(\hat{\mathbf{x}}^*_\gamma) + \gamma cp(\mathbf{x}^*_\gamma), \tag{25}$$

where inequality $(i)$ follows from the convexity of $p(\mathbf{x})$.

Inequality (25) demonstrates that

$$\gamma(1-c)p(\mathbf{x}^*_\gamma) \leq F(\hat{\mathbf{x}}^*_\gamma) - F(\mathbf{x}^*_\gamma) \leq l\|\hat{\mathbf{x}}^*_\gamma - \mathbf{x}^*_\gamma\| = l(1-c)\|\mathbf{x}^*_\gamma - \bar{\mathbf{x}}^*_\gamma\| \leq l(1-c)(\rho p(\mathbf{x}^*_\gamma))^{\frac{1}{\alpha}},$$

where the second inequality follows from the $l$-Lipschitz continuity of $F$ on $\mathcal{B}(\mathbf{x}^*_\gamma, r)$. Therefore, it holds that

$$\gamma p(\mathbf{x}^*_\gamma) \leq l(\rho p(\mathbf{x}^*_\gamma))^{\frac{1}{\alpha}}. \tag{26}$$

- **Case of $\alpha > 1$.** By (26), we have $p(\mathbf{x}^*_\gamma) \leq (\frac{\rho l^\alpha}{\gamma^\alpha})^{\frac{1}{\alpha-1}}$, which demonstrates that $p(\mathbf{x}^*_\gamma) \leq \epsilon$ if $\gamma \geq (\frac{\rho l^\alpha}{\epsilon^{\alpha-1}})^{\frac{1}{\alpha}}$.

  Then, for any $\mathbf{x}_\gamma \in \mathcal{B}(\mathbf{x}^*_\gamma, r)$ that also satisfies $p(\mathbf{x}_\gamma) \leq p(\mathbf{x}^*_\gamma) \leq \epsilon$, we have

  $$F(\mathbf{x}^*_\gamma) + \gamma p(\mathbf{x}^*_\gamma) \leq F(\mathbf{x}_\gamma) + \gamma p(\mathbf{x}_\gamma), \tag{27}$$

  which implies that $F(\mathbf{x}^*_\gamma) - F(\mathbf{x}_\gamma) \leq \gamma(p(\mathbf{x}_\gamma) - p(\mathbf{x}^*_\gamma)) \leq 0$. The desired result follows.

- **Case of $\alpha = 1$.** By (26), we have $p(\mathbf{x}^*_\gamma) = 0$ if $\gamma > \rho l$. Therefore, for any $\mathbf{x}_\gamma \in \mathcal{B}(\mathbf{x}^*_\gamma, r) \bigcap X_{\mathrm{opt}}$, by the definition of $\mathbf{x}^*_\gamma$, it holds that

  $$F(\mathbf{x}^*_\gamma) + \gamma p(\mathbf{x}^*_\gamma) \leq F(\mathbf{x}_\gamma) + \gamma p(\mathbf{x}_\gamma),$$

  which demonstrates that $F(\mathbf{x}^*_\gamma) \leq F(\mathbf{x}_\gamma)$. The desired result follows.

□

## E.7   Proof of Theorem 3.3

*Proof.* From [Beck, 2017, Theorem 10.34], the objective value after $K$ iterations can be bounded by

$$\Phi_\gamma(\mathbf{x}_K) - \Phi^*_\gamma \leq \frac{2L_\gamma\|\mathbf{x}_0 - \mathbf{x}^*\|^2}{(K+1)^2},$$

where $L_\gamma = L_{f_1} + \gamma L_{g_1}$.

Combining this with our stopping criterion, we find that after $K$ iterations,

$$\Phi_\gamma(\mathbf{x}_K) - \Phi^*_\gamma \leq \epsilon.$$

This indicates that we obtain an $\epsilon$-optimal solution to problem (P$_\gamma$). The value of $K$ satisfies:

$$K = \sqrt{\frac{2(L_{f_1} + \gamma L_{g_1})}{\epsilon}} R - 1.$$

Specifically, we analyze the value of $K$ in various scenarios in the form of $\mathcal{O}(\cdot)$.

- **Case of $\alpha > 1$.** In this case, $\gamma = \gamma^* + 2l_F^\beta \epsilon^{1-\beta}$ comprises two components: $\gamma^*$ and $2l_F^\beta \epsilon^{1-\beta}$. Therefore, it is natural to discuss which of these two components plays the dominant role in the complexity results. First, we write $K$ in the form:

  $$K = \sqrt{\frac{2(L_{f_1} + (\rho l_F^\alpha(\alpha-1)^{\alpha-1}\alpha^{-\alpha}\epsilon^{1-\alpha} + 2l_F^\beta \epsilon^{1-\beta})L_{g_1})}{\epsilon}} R - 1.$$

If $\beta < \alpha$, the dominating term in $\gamma$ is $\gamma^* = \rho l_F^\alpha (\alpha-1)^{\alpha-1} \alpha^{-\alpha} \epsilon^{1-\alpha}$. Then, the number of iterations is

$$K = \mathcal{O}\left(\sqrt{\frac{L_{f_1} + l_F^\alpha \epsilon^{1-\alpha} L_{g_1}}{\epsilon}}\right) = \mathcal{O}\left(\sqrt{\frac{L_{f_1}}{\epsilon}} + \sqrt{\frac{l_F^\alpha L_{g_1}}{\epsilon^\alpha}}\right).$$

If $\beta = \alpha$, we have $\gamma = \left(\rho(\alpha-1)^{\alpha-1}\alpha^{-\alpha} + 2\right) l_F^\alpha \epsilon^{1-\alpha}$. Then, the number of iterations is

$$K = \mathcal{O}\left(\sqrt{\frac{L_{f_1} + l_F^\alpha \epsilon^{1-\alpha} L_{g_1}}{\epsilon}}\right) = \mathcal{O}\left(\sqrt{\frac{L_{f_1}}{\epsilon}} + \sqrt{\frac{l_F^\alpha L_{g_1}}{\epsilon^\alpha}}\right).$$

If $\beta > \alpha$, the dominating term in $\gamma$ is $2l_F^\beta \epsilon^{1-\beta}$. Then, the number of iterations is

$$K = \mathcal{O}\left(\sqrt{\frac{L_{f_1} + 2l_F^\beta \epsilon^{1-\beta} L_{g_1}}{\epsilon}}\right) = \mathcal{O}\left(\sqrt{\frac{L_{f_1}}{\epsilon}} + \sqrt{\frac{l_F^\beta L_{g_1}}{\epsilon^\beta}}\right).$$

- **Case of $\alpha = 1$.** In this case, $\gamma = \gamma^* + l_F^\beta \epsilon^{1-\beta}$, where $\gamma^* = \rho l_F$. Similarly, we explore which of these two elements plays a more significant role.

  If $\beta < 1$, the dominating term in $\gamma$ is $\gamma^*$. Then, the number of iterations is

  $$K = \mathcal{O}\left(\sqrt{\frac{L_{f_1} + \rho l_F L_{g_1}}{\epsilon}}\right) = \mathcal{O}\left(\sqrt{\frac{L_{f_1}}{\epsilon}} + \sqrt{\frac{l_F L_{g_1}}{\epsilon}}\right).$$

  If $\beta = 1$, we have $\gamma = (\rho + 1)l_F \epsilon^{1-\alpha}$. Then the number of iterations is

  $$K = \mathcal{O}\left(\sqrt{\frac{L_{f_1} + (\rho + 1)l_F L_{g_1}}{\epsilon}}\right) = \mathcal{O}\left(\sqrt{\frac{L_{f_1}}{\epsilon}} + \sqrt{\frac{l_F L_{g_1}}{\epsilon}}\right).$$

  If $\beta > 1$, the dominating term in $\gamma$ is $l_F^\beta \epsilon^{1-\beta}$. Then, the number of iterations is

  $$K = \mathcal{O}\left(\sqrt{\frac{L_{f_1} + l_F^\beta \epsilon^{1-\beta} L_{g_1}}{\epsilon}}\right) = \mathcal{O}\left(\sqrt{\frac{L_{f_1}}{\epsilon}} + \sqrt{\frac{l_F^\beta L_{g_1}}{\epsilon^\beta}}\right).$$

Combining the above results, we conclude that

$$K = \mathcal{O}\left(\sqrt{\frac{L_{f_1}}{\epsilon}} + \sqrt{\frac{l_F^{\max\{\alpha,\beta\}} L_{g_1}}{\epsilon^{\max\{\alpha,\beta\}}}}\right).$$

$\square$

## E.8 Proof of Theorem 3.5

*Proof.* In this proof, we denote $\Phi_k^*$ as the optimal value of problem ($P_\gamma$) when $\gamma = \gamma_k$, and $\mathbf{x}_k$ as the output of PB-APG (Algorithm 1) in the $k$-th iteration.

- **Case of $\alpha > 1$.** Suppose that $N$ is the smallest nonnegative integer such that $\gamma_N \geq \gamma_N^* := \rho l_F^\alpha (\alpha-1)^{\alpha-1}\alpha^{-\alpha}\epsilon_N^{1-\alpha}$. In this case, we have

$$\gamma_N = \gamma_0 \nu^N \geq \rho l_F^\alpha (\alpha-1)^{\alpha-1}\alpha^{-\alpha}\epsilon_N^{1-\alpha} = \rho l_F^\alpha (\alpha-1)^{\alpha-1}\alpha^{-\alpha}\epsilon_0^{1-\alpha}(1/\eta)^{(1-\alpha)N}, \qquad (28)$$

which is equivalent to

$$\gamma_0 \left(\nu\eta^{1-\alpha}\right)^N \geq \rho l_F^\alpha (\alpha-1)^{\alpha-1}\alpha^{-\alpha}\epsilon_0^{1-\alpha}. \qquad (29)$$

From (29), after at most $N := \lceil \log_{\eta^{1-\alpha}\nu} \left(\frac{\rho l_F^\alpha (\alpha-1)^{\alpha-1}\alpha^{-\alpha}\epsilon_0^{1-\alpha}}{\gamma_0}\right) \rceil_+$ iterations, (28) holds.

Since $x_N = \text{PB-APG}(\phi_N, \psi_N, L_{f_1}, L_{g_1}, \mathbf{x}_{N-1}, \epsilon_N)$, we have

$$\Phi_N(\mathbf{x}_N) - \Phi_N^* \leq \epsilon_N, \quad \gamma_N \geq \gamma_N^*,$$

which shows that $\mathbf{x}_N$ is an $\epsilon_N$-optimal solution of ($P_\gamma$) with $\gamma = \gamma_N$. From the proof in Theorem 2.5 (see inequalities (17) and (18) in Appendix E.3), $\mathbf{x}_N$ is also an $(\frac{\epsilon_0}{\eta^N}, \frac{2\epsilon_0}{\eta^N(\gamma_0\nu^N-\gamma_N^*)})$-optimal solution of problem (P).

Furthermore, note that for any iteration $k \geq N$, inequality (29) always holds, which means that the following statement holds for any $k \geq N$:

$$\Phi_k(\mathbf{x}_k) - \Phi_k^* \leq \epsilon_k, \quad \gamma_k \geq \gamma_k^*. \tag{30}$$

Let $I_k$ be the number of iterations of PB-APG required to satisfy (30) at the $k$-th iteration of aPB-APG. Then, for any $k \geq N$, the total number of iterations is

$$K = I_0 + I_1 + \cdots + I_k.$$

From [Beck, 2017, Theorem 10.34], the number of iterations in $i$-th inner loop satisfies:

$$I_i = \sqrt{\frac{2(L_{f_1} + \gamma_i L_{g_1})}{\epsilon_i}}\|\mathbf{x}_{i-1} - \mathbf{x}_i^*\| - 1,$$

where $\mathbf{x}_i^*$ is the optimal solution in $i$-th inner loop. Then we have that

$$
\begin{aligned}
K &= \sum_{i=0}^{k} \sqrt{\frac{2(L_{f_1} + \gamma_i L_{g_1})}{\epsilon_i}}\|\mathbf{x}_{i-1} - \mathbf{x}_i^*\| - k \\
&\leq \sum_{i=0}^{k} \sqrt{\frac{2(L_{f_1} + \gamma_k L_{g_1})}{\epsilon_i}}R - k \\
&= \frac{\eta^{\frac{k}{2}} - 1}{\eta^{\frac{1}{2}} - 1}\sqrt{\frac{2(L_{f_1} + \gamma_0\nu^k L_{g_1})}{\epsilon_0}} - k.
\end{aligned}
$$

For simplicity, we can also use $\mathcal{O}(\cdot)$ to show the value of $K$.

$$
\begin{aligned}
K &= \mathcal{O}\left(\sqrt{\frac{L_{f_1} + \gamma_0 L_{g_1}}{\epsilon_0}}\right) + \cdots + \mathcal{O}\left(\sqrt{\frac{L_{f_1} + \gamma_k L_{g_1}}{\epsilon_k}}\right) \\
&\leq \mathcal{O}\left(\sqrt{\frac{L_{f_1} + \gamma_k L_{g_1}}{\epsilon_0}}\right) + \cdots + \mathcal{O}\left(\sqrt{\frac{L_{f_1} + \gamma_k L_{g_1}}{\epsilon_k}}\right) \\
&= \mathcal{O}\left(\sqrt{\frac{L_{f_1} + \gamma_k L_{g_1}}{\epsilon_k}}\left(1 + \sqrt{1/\eta} + \sqrt{1/\eta^2} + \cdots + \sqrt{1/\eta^k}\right)\right) \\
&= \mathcal{O}\left(\sqrt{\frac{L_{f_1} + \gamma_k L_{g_1}}{\epsilon_k}}\right) \\
&= \mathcal{O}\left(\sqrt{\frac{L_{f_1}\eta^k}{\epsilon_0}} + \sqrt{\frac{L_{g_1}\gamma_0(\eta\nu)^k}{\epsilon_0}}\right).
\end{aligned}
$$

- **Case of $\alpha = 1$.** Suppose that after $N$ updates, we have $\gamma_N \geq \rho l_F$, i.e.,

$$\gamma_0\nu^N \geq \rho l_F. \tag{31}$$

This demonstrates that after for all $k \geq N := \log_\nu\left(\frac{\rho l_F}{\gamma_0}\right)$, (31) always holds.

Similar to the case of $\alpha > 1$, the total iteration number is:

$$
\begin{aligned}
K &= \mathcal{O}\left(\sqrt{\frac{L_{f_1} + \gamma_0 L_{g_1}}{\epsilon_0}}\right) + \cdots + \mathcal{O}\left(\sqrt{\frac{L_{f_1} + \gamma_k L_{g_1}}{\epsilon_k}}\right) \\
&= \mathcal{O}\left(\sqrt{\frac{L_{f_1} + \gamma_k L_{g_1}}{\epsilon_k}}\right) \\
&= \mathcal{O}\left(\sqrt{\frac{L_{f_1}\eta^k}{\epsilon_0}} + \sqrt{\frac{L_{g_1}\gamma_0(\eta\nu)^k}{\epsilon_0}}\right).
\end{aligned}
$$

$\square$

### E.9 Proof of Theorem 3.8

Before proving Theorem 3.8, we need the following lemma that is modified from Theorem 1 in Lin and Xiao [2014], we state it in the subsequent lemma for completeness.

**Lemma E.1.** *Suppose that Assumptions 2.1, 3.1, 3.2, and 3.7 hold. Let $\mathbf{x}_\gamma^*$ be an optimal solution of problem $(\mathrm{P}_\gamma)$ and suppose that there exists a constant $R$ such that $\max\{\|\mathbf{y}_0 - \mathbf{x}_\gamma^*\|, \|\tilde{\mathbf{x}} - \mathbf{x}_\gamma^*\|\} \leq R$. Then, the sequence $\{\mathbf{x}_k\}$ generated by Algorithm 3 satisfy*

$$\Phi_\gamma(\mathbf{x}_k) - \Phi_\gamma(\mathbf{x}_\gamma^*) \leq \left(\frac{L_\gamma + \mu}{2} R^2\right) \left(1 - \sqrt{\frac{\mu}{L_\gamma}}\right)^k. \tag{32}$$

*Proof.* Denote $L_\gamma = L_{f_1} + \gamma L_{g_1}$. By Theorem 3.1 in Beck and Teboulle [2009], we have

$$\Phi_\gamma(\tilde{\mathbf{x}}) - \Phi_\gamma(\mathbf{x}_\gamma^*) \leq \frac{L_\gamma}{2} \|\mathbf{y}_0 - \mathbf{x}_\gamma^*\|^2. \tag{33}$$

Utilize Theorem 1 in Lin and Xiao [2014], we have

$$
\begin{aligned}
\Phi_\gamma(\mathbf{x}_k) - \Phi_\gamma(\mathbf{x}_\gamma^*) &\leq \left(\Phi_\gamma(\tilde{\mathbf{x}}) - \Phi_\gamma(\mathbf{x}_\gamma^*) + \frac{\mu}{2}\|\tilde{\mathbf{x}} - \mathbf{x}_\gamma^*\|^2\right) \left(1 - \sqrt{\frac{\mu}{L_\gamma}}\right)^k \\
&\overset{(33)}{\leq} \left(\frac{L_\gamma}{2}\|\mathbf{y}_0 - \mathbf{x}_\gamma^*\|^2 + \frac{\mu}{2}\|\tilde{\mathbf{x}} - \mathbf{x}_\gamma^*\|^2\right) \left(1 - \sqrt{\frac{\mu}{L_\gamma}}\right)^k \\
&\leq \left(\frac{L_\gamma + \mu}{2} R^2\right) \left(1 - \sqrt{\frac{\mu}{L_\gamma}}\right)^k.
\end{aligned}
\tag{34}
$$

$\square$

By Lemma E.1, we are now prepared to prove Theorem 3.8.

*Proof.* By Lemma E.1, the number of iterations required to achieve an $\epsilon$-optimal solution for problem $(\mathrm{P}_\gamma)$ is

$$K = \mathcal{O}\left(\sqrt{\frac{L_\gamma}{\mu}} \log\left(\frac{L_\gamma + \mu}{2\epsilon} R^2\right)\right) = \mathcal{O}\left(\sqrt{\frac{L_\gamma}{\mu}} \log\frac{1}{\epsilon}\right).$$

- **Case of $\alpha > 1$.** In this case, $\gamma = \gamma^* + 2l_F^\beta \epsilon^{1-\beta}$, where $\gamma^* = \rho l_F^\alpha (\alpha - 1)^{\alpha-1} \alpha^{-\alpha} \epsilon^{1-\alpha}$.

  If $\beta < \alpha$, the dominating term in $\gamma$ is $\gamma^*$. Then, the number of iterations is

  $$K = \mathcal{O}\left(\sqrt{\frac{L_{f_1} + l_F^\alpha \epsilon^{1-\alpha} L_{g_1}}{\mu}} \log\frac{1}{\epsilon}\right) = \mathcal{O}\left(\sqrt{\frac{L_{f_1}}{\mu}} \log\frac{1}{\epsilon} + \sqrt{\frac{l_F^\alpha L_{g_1}}{\epsilon^{\alpha-1}}} \log\frac{1}{\epsilon}\right).$$

  If $\beta = \alpha$, we have $\gamma = \left(\rho(\alpha - 1)^{\alpha-1}\alpha^{-\alpha} + 2\right) l_F^\alpha \epsilon^{1-\alpha}$. Then, the number of iterations is

  $$K = \mathcal{O}\left(\sqrt{\frac{L_{f_1} + l_F^\alpha \epsilon^{1-\alpha} L_{g_1}}{\mu}} \log\frac{1}{\epsilon}\right) = \mathcal{O}\left(\sqrt{\frac{L_{f_1}}{\mu}} \log\frac{1}{\epsilon} + \sqrt{\frac{l_F^\alpha L_{g_1}}{\epsilon^{\alpha-1}}} \log\frac{1}{\epsilon}\right).$$

  If $\beta > \alpha$, the dominating term in $\gamma$ is $2l_F^\beta \epsilon^{1-\beta}$. Then, the number of iterations is

  $$K = \mathcal{O}\left(\sqrt{\frac{L_{f_1} + 2l_F^\beta \epsilon^{1-\beta} L_{g_1}}{\mu}} \log\frac{1}{\epsilon}\right) = \mathcal{O}\left(\sqrt{\frac{L_{f_1}}{\mu}} \log\frac{1}{\epsilon} + \sqrt{\frac{l_F^\beta L_{g_1}}{\epsilon^{\beta-1}}} \log\frac{1}{\epsilon}\right).$$

- **Case of $\alpha = 1$.** When $\alpha = 1$, $\gamma$ can be written as $\gamma = \gamma^* + l_F^\beta \epsilon^{1-\beta}$, where $\gamma^* = \rho l_F$.

  If $\beta < 1$, the dominating term in $\gamma$ is $\gamma^*$. Then, the number of iterations is

  $$K = \mathcal{O}\left(\sqrt{\frac{L_{f_1} + \rho l_F L_{g_1}}{\mu}} \log\frac{1}{\epsilon}\right) = \mathcal{O}\left(\sqrt{\frac{L_{f_1}}{\mu}} \log\frac{1}{\epsilon} + \sqrt{\frac{l_F L_{g_1}}{\epsilon^{\alpha-1}}} \log\frac{1}{\epsilon}\right).$$

  If $\beta = 1$, we have $\gamma = (\rho + 1)l_F \epsilon^{1-\alpha}$. Then, the number of iterations is

  $$K = \mathcal{O}\left(\sqrt{\frac{L_{f_1} + \rho l_F L_{g_1}}{\mu}} \log\frac{1}{\epsilon}\right) = \mathcal{O}\left(\sqrt{\frac{L_{f_1}}{\mu}} \log\frac{1}{\epsilon} + \sqrt{\frac{l_F L_{g_1}}{\epsilon^{\alpha-1}}} \log\frac{1}{\epsilon}\right).$$

If $\beta > 1$, the dominating term in $\gamma$ is $l_F^\beta \epsilon^{1-\beta}$. Then, we have

$$K = \mathcal{O}\left(\sqrt{\frac{L_{f_1} + l_F^\beta \epsilon^{1-\beta} L_{g_1}}{\mu}} \log \frac{1}{\epsilon}\right) = \mathcal{O}\left(\sqrt{\frac{L_{f_1}}{\mu}} \log \frac{1}{\epsilon} + \sqrt{\frac{l_F^\beta L_{g_1}}{\epsilon^{\beta-1}}} \log \frac{1}{\epsilon}\right).$$

Combining the above results, we conclude that

$$K = \mathcal{O}\left(\sqrt{\frac{L_{f_1}}{\mu}} \log \frac{1}{\epsilon} + \sqrt{\frac{l_F^{\max\{\alpha,\beta\}} L_{g_1}}{\epsilon^{\max\{\alpha-1,\beta-1\}}}} \log \frac{1}{\epsilon}\right).$$

$\square$

## E.10 Proof of Theorem 3.9

*Proof.* Denote $l_\gamma = l_{f_2} + \gamma l_{g_2}$. Define $\Phi_{\gamma,best}^K = \min_{i=0,...,K} \Phi_\gamma(\mathbf{x}_i)$ and $\hat{\Phi}_{\gamma,best}^{K,j} = \min_{i=j,...,K} \Phi_\gamma(\mathbf{x}_i)$ for all $0 \le j \le K$. We claim that the sequence generated by the subgradient method satisfies

$$\Phi_{\gamma,best}^K - \Phi_\gamma^* \le \frac{l_\gamma}{4} \frac{R^2 + 2\log 2}{\sqrt{K+2}}. \tag{35}$$

Specifically, from Lemma 8.24 in Beck [2017], for all $0 \le j \le K$, we have

$$\hat{\Phi}_{\gamma,best}^{K,j} - \Phi_\gamma^* \le \frac{1}{2} \frac{R^2 + \sum_{k=j}^K \eta_k^2 \|\xi_k\|^2}{\sum_{k=j}^K \eta_k}. \tag{36}$$

Define $\lfloor \cdot \rfloor$ and $\lceil \cdot \rceil$ as rounding up and rounding down, respectively. Let $j = \lfloor \frac{K}{2} \rfloor$ in (36), by the definition of step-size $\eta_k = \frac{R}{l_\gamma \sqrt{k+1}}$, we have

$$\hat{\Phi}_{\gamma,best}^{K,j} - \Phi_\gamma^* \le \frac{l_\gamma}{2} \frac{R^2 + \sum_{k=\lfloor \frac{K}{2} \rfloor}^K \frac{1}{k+1}}{\sum_{k=\lfloor \frac{K}{2} \rfloor}^K \frac{1}{\sqrt{k+1}}} \le \frac{l_\gamma}{4} \frac{R^2 + 2\log 2}{\sqrt{K+2}}, \tag{37}$$

where the second inequality follows from that $\sum_{k=\lfloor \frac{K}{2} \rfloor}^K \frac{1}{k+1} \le \int_{\lceil \frac{K}{2} \rceil - 1}^K \frac{1}{s+1} ds \le 2\log 2$ and $\sum_{k=\lfloor \frac{K}{2} \rfloor}^K \frac{1}{\sqrt{k+1}} \ge \int_{\lceil \frac{K}{2} \rceil}^{K+1} \frac{1}{\sqrt{s+1}} ds \ge \frac{1}{2}\sqrt{K+2}$.

From the fact that $\Phi_{\gamma,best}^K \le \hat{\Phi}_{\gamma,best}^{K,j}$, The desired result of (35) follows.

Then, inequality (35) demonstrates that the number of iterations to obtain an $\epsilon$-optimal solution for problem $(\mathrm{P}_\gamma)$ is

$$K = \mathcal{O}\left(\frac{l_{f_2} + \gamma l_{g_2}}{\epsilon}\right)^2.$$

- **Case of $\alpha > 1$.** we have $\gamma = \gamma^* + 2l_{f_2}^\beta \epsilon^{1-\beta}$ and $\gamma^* = \rho l_{f_2}^\alpha (\alpha - 1)^{\alpha-1} \alpha^{-\alpha} \epsilon^{1-\alpha}$.

  If $\beta < \alpha$, the dominating term in $\gamma$ is $\gamma^*$. Then, the number of iterations is

  $$K = \mathcal{O}\left(\frac{l_{f_2} + l_{f_2}^\alpha \epsilon^{1-\alpha} l_{g_2}}{\epsilon}\right)^2 = \mathcal{O}\left(\frac{l_{f_2}^2}{\epsilon^2} + \frac{l_{f_2}^{2\alpha} l_{g_2}^2}{\epsilon^{2\alpha}}\right).$$

  If $\beta = \alpha$, we have $\gamma = \left(\rho(\alpha - 1)^{\alpha-1} \alpha^{-\alpha} + 2\right) l_F^\alpha \epsilon^{1-\alpha}$. Then, the number of iterations is

  $$K = \mathcal{O}\left(\frac{l_{f_2} + l_{f_2}^\alpha \epsilon^{1-\alpha} l_{g_2}}{\epsilon}\right)^2 = \mathcal{O}\left(\frac{l_{f_2}^2}{\epsilon^2} + \frac{l_{f_2}^{2\alpha} l_{g_2}^2}{\epsilon^{2\alpha}}\right).$$

  If $\beta > \alpha$, the dominating term in $\gamma$ is $2l_F^\beta \epsilon^{1-\beta}$. Then, the number of iterations is

  $$K = \mathcal{O}\left(\frac{l_{f_2} + 2l_{f_2}^\beta \epsilon^{1-\beta} l_{g_2}}{\epsilon}\right)^2 = \mathcal{O}\left(\frac{l_{f_2}^2}{\epsilon^2} + \frac{l_{f_2}^{2\beta} l_{g_2}^2}{\epsilon^{2\beta}}\right).$$

- **Case of $\alpha = 1$.** we have $\gamma = \gamma^* + l_{f_2}^\beta \epsilon^{1-\beta}$ and $\gamma^* = \rho l_{f_2}$.

  If $\beta < 1$, the dominating term in $\gamma$ is $\gamma^*$. Then, the number of iterations is

$$K = \mathcal{O}\left(\frac{l_{f_2} + \rho l_{f_2} l_{g_2}}{\epsilon}\right)^2 = \mathcal{O}\left(\frac{l_{f_2}^2}{\epsilon^2} + \frac{l_{f_2}^2 l_{g_2}^2}{\epsilon^2}\right).$$

  If $\beta = 1$, we have $\gamma = (\rho + 1)l_F \epsilon^{1-\alpha}$. Then, the number of iterations is

$$K = \mathcal{O}\left(\frac{l_{f_2} + \rho l_{f_2} l_{g_2}}{\epsilon}\right)^2 = \mathcal{O}\left(\frac{l_{f_2}^2}{\epsilon^2} + \frac{l_{f_2}^2 l_{g_2}^2}{\epsilon^2}\right).$$

  If $\beta > 1$, the dominating term in $\gamma$ is $l_{f_2}^\beta \epsilon^{1-\beta}$. Then, the number of iterations is

$$K = \mathcal{O}\left(\frac{l_{f_2} + l_{f_2}^\beta l_{g_2} \epsilon^{1-\beta}}{\epsilon}\right)^2 = \mathcal{O}\left(\frac{l_{f_2}^2}{\epsilon^2} + \frac{l_{f_2}^{2\beta} l_{g_2}^2}{\epsilon^{2\beta}}\right).$$

Combining the above results, we conclude that

$$K = \mathcal{O}\left(\frac{l_{f_2}^2}{\epsilon^2} + \frac{l_{f_2}^{\max\{2\alpha, 2\beta\}} l_{g_2}^2}{\epsilon^{\max\{2\alpha, 2\beta\}}}\right).$$

$\square$

## E.11 Proof of Theorem 3.10

*Proof.* Denote $l_\gamma = l_{f_2} + \gamma l_{g_2}$, define $\Phi_{\gamma, best}^K = \min\limits_{i=0,\ldots,K} \Phi_\gamma(\mathbf{x}_i)$. From Theorem 8.31 in Beck [2017], the sequence generated by the subgradient method satisfies

$$\Phi_{\gamma, best}^K - \Phi_\gamma^* \le \frac{2l_\gamma^2}{\mu_{f_2}(K+1)}.$$

This demonstrates that the number of iterations to obtain an $\epsilon$-optimal solution for problem $(\mathrm{P}_\gamma)$ is

$$K = \mathcal{O}\left(\frac{(l_{f_2} + \gamma l_{g_2})^2}{\mu_{f_2}\epsilon}\right).$$

- **Case of $\alpha > 1$.** we have $\gamma = \gamma^* + 2l_{f_2}^\beta \epsilon^{1-\beta}$ and $\gamma^* = \rho l_{f_2}^\alpha (\alpha - 1)^{\alpha-1} \alpha^{-\alpha} \epsilon^{1-\alpha}$.

  If $\beta < \alpha$, the dominating term in $\gamma$ is $\gamma^*$. Then, the number of iterations is

$$K = \mathcal{O}\left(\frac{(l_{f_2} + l_{f_2}^\alpha \epsilon^{1-\alpha} l_{g_2})^2}{\mu_{f_2}\epsilon}\right) = \mathcal{O}\left(\frac{l_{f_2}^2}{\mu_{f_2}\epsilon} + \frac{l_{f_2}^{2\alpha} l_{g_2}^2}{\mu_{f_2}\epsilon^{2\alpha-1}}\right).$$

  If $\beta = \alpha$, we have $\gamma = \left(\rho(\alpha-1)^{\alpha-1}\alpha^{-\alpha} + 2\right) l_F^\alpha \epsilon^{1-\alpha}$. Then, the number of iterations is

$$K = \mathcal{O}\left(\frac{(l_{f_2} + l_{f_2}^\alpha \epsilon^{1-\alpha} l_{g_2})^2}{\mu_{f_2}\epsilon}\right) = \mathcal{O}\left(\frac{l_{f_2}^2}{\mu_{f_2}\epsilon} + \frac{l_{f_2}^{2\alpha} l_{g_2}^2}{\mu_{f_2}\epsilon^{2\alpha-1}}\right).$$

  If $\beta > \alpha$, the dominating term in $\gamma$ is $2l_F^\beta \epsilon^{1-\beta}$. Then, the number of iterations is

$$K = \mathcal{O}\left(\frac{(l_{f_2} + 2l_{f_2}^\beta \epsilon^{1-\beta} l_{g_2})^2}{\mu_{f_2}\epsilon}\right) = \mathcal{O}\left(\frac{l_{f_2}^2}{\mu_{f_2}\epsilon} + \frac{l_{f_2}^{2\beta} l_{g_2}^2}{\mu_{f_2}\epsilon^{2\beta-1}}\right).$$

- **Case of $\alpha > 1$.** we have $\gamma = \gamma^* + l_{f_2}^\beta \epsilon^{1-\beta}$ and $\gamma^* = \rho l_{f_2}$.

  If $\beta < 1$, the dominating term in $\gamma$ is $\gamma^*$. Then, the number of iterations is

$$K = \mathcal{O}\left(\frac{(l_{f_2} + \rho l_{f_2} l_{g_2})^2}{\mu_{f_2}\epsilon}\right) = \mathcal{O}\left(\frac{l_{f_2}^2}{\mu_{f_2}\epsilon} + \frac{l_{f_2}^2 l_{g_2}^2}{\mu_{f_2}\epsilon}\right).$$

If $\beta = 1$, we have $\gamma = (\rho + 1)l_F \epsilon^{1-\alpha}$. Then, the number of iterations is

$$K = \mathcal{O}\left(\frac{(l_{f_2} + \rho l_{f_2} l_{g_2})^2}{\mu_{f_2} \epsilon}\right) = \mathcal{O}\left(\frac{l_{f_2}^2}{\mu_{f_2} \epsilon} + \frac{l_{f_2}^2 l_{g_2}^2}{\mu_{f_2} \epsilon}\right).$$

If $\beta > 1$, the dominating term in $\gamma$ is $l_{f_2}^\beta \epsilon^{1-\beta}$. Then, the number of iterations is

$$K = \mathcal{O}\left(\frac{(l_{f_2} + l_{f_2}^\beta l_{g_2} \epsilon^{1-\beta})^2}{\mu_{f_2} \epsilon}\right) = \mathcal{O}\left(\frac{l_{f_2}^2}{\mu_{f_2} \epsilon} + \frac{l_{f_2}^{2\beta} l_{g_2}^2}{\mu_{f_2} \epsilon^{2\beta-1}}\right).$$

Combining the above results, we conclude that

$$K = \mathcal{O}\left(\frac{l_{f_2}^2}{\mu_{f_2} \epsilon} + \frac{l_{f_2}^{\max\{2\alpha, 2\beta\}} l_{g_2}^2}{\mu_{f_2} \epsilon^{\max\{2\alpha-1, 2\beta-1\}}}\right).$$

$\square$

# F  Implementation details

In this section, we provide supplementary experiment settings and results. Specifically, in Appendix F.1, we present the detailed experimental settings, and in Appendix F.2, we provide the detailed experimental results. Additionally, in Appendix F.3 and F.4, we conduct experiments with different values of penalty parameter $\gamma$ and solution accuracy $\epsilon$, respectively.

## F.1  Experiment setting

All simulations are implemented using MATLAB R2023a on a PC running Windows 11 with an AMD (R) Ryzen (TM) R7-7840H CPU (3.80GHz) and 16GB RAM.

### F.1.1  Experiment setting of Section 4.1

We conduct the first experiment using the `a1a.t` data from LIBSVM datasets[5]. This data consists of $30,956$ instances, each with $n = 123$ features. For this experiment, a sample of $1,000$ instances is taken from the data, denoted as $A$. The corresponding labels for these instances are denoted as $b$, where each label $b_i$ is either $-1$ or $1$, corresponding to the $i$-th instance $\mathbf{a}_i$.

The Greedy FISTA algorithm [Liang et al., 2022] is used as a benchmark to compute $G^*$. To compute the proximal mapping of $f_2(\mathbf{x}) + \gamma g_2(\mathbf{x})$ in problem $(P_\gamma)$, i.e, projection onto a 1-norm ball, we utilize the method proposed in Duchi et al. [2008], which performs exact projection in $\mathcal{O}(n)$ expected time, where n is the dimension of $\mathbf{x}$.

For the PB-APG and PB-APG-sc algorithms, we set the value of $\gamma = 10^5$, and we terminate the algorithms when $\|\mathbf{x}_{k+1} - \mathbf{x}_k\| \leq 10^{-10}$. For the aPB-APG and aPB-APG-sc algorithms, we set $\gamma_0 = \frac{1}{2^5}$, $\nu = 20$, $\eta = 10$, and $\epsilon_0 = 10^{-6}$. The iterations of these two algorithms continue until $\epsilon_k$ reaches $10^{-10}$ (meanwhile, $\gamma = 10^5$).

We compare our methods with MNG, BiG-SAM, DBGD, a-IRG, CG-BiO, Bi-SG, and R-APM in this experiment. Specifically, for R-APM [Samadi et al., 2023], the regularization parameter $\eta$ is set to $\eta = 1/\gamma$, reflecting the equivalence of the penalty formulation $(P_\gamma)$ to $(P_{\text{Reg}})$, with $\sigma = 1/\gamma$, as previously discussed.

We note that the termination criterion $\|\mathbf{x}_{k+1} - \mathbf{x}_k\| \leq 10^{-10}$ used in our experiments is different from the one proposed in our algorithms since the parameters required for the latter are not easily measurable. Nevertheless, this termination criterion is also widely used in the literature, as it corresponds to a gradient mapping [Beck, 2017, Nesterov, 2018, Davis and Drusvyatskiy, 2019]. Furthermore, Theorem 3.5 of Drusvyatskiy and Lewis [2018b] implies that $\|\mathbf{x}_{k+1} - \mathbf{x}_k\|$ also measures the distance to the optimal solution set.

### F.1.2  Experiment setting of Section 4.2

In the second experiment, we address the problem of least squares regression using the `YearPredictionMSD` data from the UCI Machine Learning Repository[6]. This data consists of $515,345$ songs with release years ranging from 1992 to 2011. Each song has 90 features, and the corresponding release year is used as the label.

---

[5]https://www.csie.ntu.edu.tw/~cjlin/libsvmtools/datasets/binary/a1a.t
[6]https://archive.ics.uci.edu/ml/datasets/YearPredictionMSD

For this experiment, a sample of $m = 1,000$ songs is taken from the data, and the feature matrix and release years vector are denoted as $A$ and $b$, respectively.

Following Section 5.2 in Merchav and Sabach [2023], we apply the min-max scaling technique to normalize the feature matrix $A$. Additionally, we add an intercept term and 90 collinear features to $A$ such that the resulting matrix $A^T A$ becomes positive semi-definite, which implies that the feasible set $X_{\mathrm{opt}}$ is not a singleton.

We compare our methods with a-IRG, BiG-SAM, and Bi-SG in this experiment. Specifically, for BiG-SAM [Sabach and Shtern, 2017], we consider the accuracy parameter $\delta$ for the Moreau envelope with two values, namely $\delta = 1$ and $\delta = 0.01$.

To benchmark the performance, we utilize the MATLAB function `lsqminnorm` to compute $G^*$. Moreover, we follow the parameter settings outlined in Section 4.1.

## F.2  Detailed results of experiments

To approximate the optimal value $F^*$, we use the MATLAB function `fmincon` to solve a relaxed version of the function-value-based reformulations in equation $(\mathrm{P_{Val}})$. In this relaxed version, we replace the constraint in $(\mathrm{P_{Val}})$ with $G(\mathbf{x}) - G^* \leq \varepsilon$, where $\varepsilon = 10^{-10}$. This allows us to obtain an approximation of the optimal value while allowing for a small deviation from the true optimal value $G^*$.

We gather the total number of iterations for our methods, as well as the lower- and upper-level objective values and the optimal gaps for all the methods, in Table 3. Subsequently, we compare the optimal gaps of all methods, which are defined as $G(\mathbf{x}) - G^*$ and $F(\mathbf{x}) - F^*$ for the lower- and upper-level optimal gaps, respectively.

Table 3: Methods comparison: lower- and upper-level objectives and optimal gaps

| Method | Total iterations | Lower-level value | Lower-level gap | Upper-level value | Upper-level gap |
|---|---|---|---|---|---|
| Logistic Regression Problem (5) | | | | | |
| PB-APG | 1470 | 3.2794e-01 | **1.7630e-08** | 4.9382e+00 | **-3.3998e-03** |
| aPB-APG | 1010 | 3.2794e-01 | **1.7630e-08** | 4.9382e+00 | **-3.3998e-03** |
| PB-APG-sc | 2278 | 3.2794e-01 | **1.7630e-08** | 4.9382e+00 | **-3.3998e-03** |
| aPB-APG-sc | 1046 | 3.2794e-01 | **1.7630e-08** | 4.9382e+00 | **-3.3998e-03** |
| MNG | / | 3.4540e-01 | 1.7459e-02 | 1.7469e+00 | -3.1947e+00 |
| BiG-SAM | / | 3.3878e-01 | 1.0840e-02 | 2.2873e+00 | -2.6543e+00 |
| DBGD | / | 5.2681e-01 | 1.9887e-01 | 8.8408e+00 | -4.8532e+00 |
| a-IRG | / | 3.3765e-01 | 9.7121e-03 | 2.5401e+00 | -2.4016e+00 |
| CG-BiO | / | 4.3040e-01 | 1.0246e-01 | 3.7684e-01 | -4.5648e+00 |
| Bi-SG | / | 3.2806e-01 | 1.1530e-04 | 4.6873e+00 | -2.5432e-01 |
| R-APM | / | 3.2794e-01 | 1.7645e-08 | 4.9382e+00 | -3.4013e-03 |
| Least Squares Regression Problem (6) | | | | | |
| PB-APG | 39314 | 7.3922e-03 | 6.0034e-07 | 4.7236e+00 | -1.1888e-01 |
| aPB-APG | 40784 | 7.3922e-03 | **6.0030e-07** | 4.7236e+00 | **-1.1887e-01** |
| PB-APG-sc | 46446 | 7.3922e-03 | 6.0034e-07 | 4.7236e+00 | -1.1888e-01 |
| aPB-APG-sc | 61777 | 7.3922e-03 | 6.0035e-07 | 4.7236e+00 | -1.1888e-01 |
| BiG-SAM ($\delta = 1$) | / | 7.5189e-03 | 1.2733e-04 | 3.5081e+00 | -1.3344e+00 |
| BiG-SAM ($\delta = 0.01$) | / | 7.3958e-03 | 4.2281e-06 | 5.8510e+01 | 5.3668e+01 |
| a-IRG | / | 1.6224e-02 | 8.8328e-03 | 4.7745e-01 | -4.3651e+00 |
| Bi-SG | / | 8.5782e-03 | 1.1866e-03 | 1.3832e+00 | -3.4593e+00 |

Table 3 reveals that for the logistic regression problem (5), our PB-APG, aPB-APG, PB-APG-sc, and aPB-APG-sc exhibit almost identical function values for both objectives, surpassing other methods in terms of optimal gaps for the lower- and upper-level objectives (measured by the numerical value of the upper-level objective). In the case of the least squares regression problem (6), aPB-APG achieves the smallest optimal gaps for both objectives, followed by PB-APG and PB-APG-sc. These results demonstrate that our methods, despite yielding larger upper-level function values, generate solutions that are significantly closer to the optimal solution, as depicted in Figure 1. Additionally, for the problem in (5), both aPB-APG and aPB-APG-sc require fewer iterations than PB-APG and PB-APG-sc, respectively. This can be attributed to the warm-start mechanism employed in aPB-APG and aPB-APG-sc. Moreover, for the problem in (6), both aPB-APG and aPB-APG-sc require more iterations than PB-APG and PB-APG-sc, respectively. However, they exhibit staircase-shaped curves, which avoid the unwanted oscillations in PB-APG and PB-APG-sc, we have a similar observation in Figure 2.

## F.3  Supplementary experiments for different penalty parameters

In this section, we investigate the impact of different values of penalty parameter $\gamma$ on the experimental results of problems (5) and (6). We set $\gamma$ to be either $2 \times 10^4$ or $5 \times 10^5$ for PB-APG and PB-APG-sc, and choose the

corresponding $\gamma_0$ values as $\frac{0.2}{2^5}$ or $\frac{5}{2^5}$ for aPB-APG and aPB-APG-sc, respectively. The remaining settings are the same as in Section 4.

We plot the values of the residuals of the lower-level objective $G(\mathbf{x}_k) - G^*$ and the upper-level objective over time in Figures 3 and 4. Additionally, we also collect the total number of iterations, the lower- and upper-level objective values, and the optimal gaps of our methods in Table 4 for problems (5) and (6) with different values of $\gamma$.

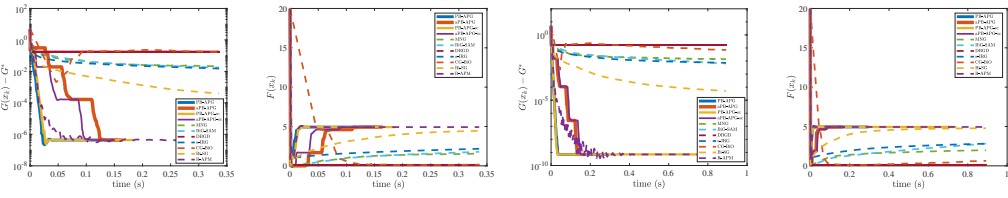

Figure 3: LRP (5) with $\gamma = 2 \times 10^4$ (left two subfigures) and $\gamma = 5 \times 10^5$ (right two subfigures).

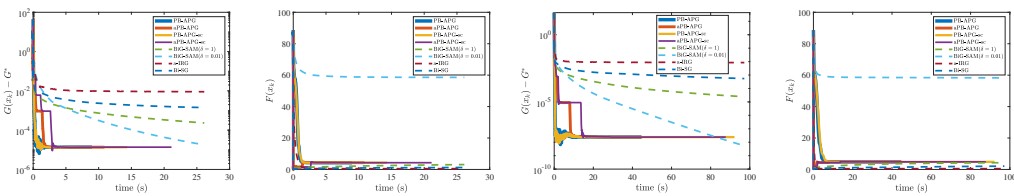

Figure 4: LSRP (6) with $\gamma = 2 \times 10^4$ (left two subfigures) and $\gamma = 5 \times 10^5$ (right two subfigures).

As Figures 3 and 4 show, our methods consistently outperform the other methods for both the lower- and upper-level objectives, irrespective of the penalty parameter $\gamma$, since our methods achieve lower optimal gaps and desired function values for the lower- and upper-level objectives, respectively. The only exception is problem (6) with $\gamma = 5 \times 10^5$, as the third subfigure of Figure 4 shows, since we do not set the solution accuracy of BiG-SAM ($\delta = 0.01$), it attains a lower optimal gap than our PB-APG-sc and aPB-APG-sc for the lower-level objective. However, BiG-SAM ($\delta = 0.01$) produces significantly worse upper-level objective values, which are much larger than the objective values of our methods.

Table 4: Lower- and upper-level objectives and optimal gaps with different penalty parameters for problem (5).

| | | $\gamma = 2 \times 10^4$ | | | |
|---|---|---|---|---|---|
| Method | Total iterations | Lower-level value | Lower-level gap | Upper-level value | Upper-level gap |
| PB-APG | 883 | 3.2794e-01 | 4.3569e-07 | 4.9243e+00 | -1.7362e-02 |
| aPB-APG | 967 | 3.2794e-01 | 4.3569e-07 | 4.9243e+00 | -1.7362e-02 |
| PB-APG-sc | 1123 | 3.2794e-01 | 4.3569e-07 | 4.9243e+00 | -1.7362e-02 |
| aPB-APG-sc | 879 | 3.2794e-01 | 4.3569e-07 | 4.9243e+00 | -1.7362e-02 |
| | | $\gamma = 5 \times 10^5$ | | | |
| Method | Total iterations | Lower-level value | Lower-level gap | Upper-level value | Upper-level gap |
| PB-APG | 1623 | 3.2794e-01 | 7.0685e-10 | 4.9410e+00 | -5.7820e-04 |
| aPB-APG | 976 | 3.2794e-01 | 7.0685e-10 | 4.9410e+00 | -5.7820e-04 |
| PB-APG-sc | 4848 | 3.2794e-01 | 7.0684e-10 | 4.9410e+00 | -5.7820e-04 |
| aPB-APG-sc | 1018 | 3.2794e-01 | 7.0687e-10 | 4.9410e+00 | -5.7821e-04 |

Tables 3, 4, and 5 reveal that the number of iterations for our methods increases as penalty parameter $\gamma$ increases. However, it is worth noting that the accuracy of the obtained solutions also increases, as indicated by the decreasing optimal gaps of the lower- and upper-level objectives. This observation confirms that the complexity results and solution accuracies of our methods are indeed dependent on the choice of penalty parameters, specifically, $L_\gamma$, as demonstrated in corresponding Theorem 3.3 and other related theorems.

Table 5: Lower- and upper-level objectives and optimal gaps with different penalty parameters for problem (6).

| | | $\gamma = 2 \times 10^4$ | | | |
|---|---|---|---|---|---|
| Method | Total iterations | Lower-level value | Lower-level gap | Upper-level value | Upper-level gap |
| PB-APG | 17153 | 7.4052e-03 | 1.3619e-05 | 4.2843e+00 | -5.5818e-01 |
| aPB-APG | 20877 | 7.4052e-03 | 1.3619e-05 | 4.2843e+00 | -5.5818e-01 |
| PB-APG-sc | 27501 | 7.4052e-03 | 1.3619e-05 | 4.2843e+00 | -5.5818e-01 |
| aPB-APG-sc | 40077 | 7.4052e-03 | 1.3619e-05 | 4.2843e+00 | -5.5818e-01 |
| | | $\gamma = 5 \times 10^5$ | | | |
| Method | Total iterations | Lower-level value | Lower-level gap | Upper-level value | Upper-level gap |
| PB-APG | 85511 | 7.3916e-03 | 2.4094e-08 | 4.8198e+00 | -2.2752e-02 |
| aPB-APG | 85502 | 7.3916e-03 | 2.4093e-08 | 4.8198e+00 | -2.2752e-02 |
| PB-APG-sc | 173731 | 7.3916e-03 | 2.4071e-08 | 4.8198e+00 | -2.2740e-02 |
| aPB-APG-sc | 166324 | 7.3916e-03 | 2.4091e-08 | 4.8198e+00 | -2.2751e-02 |

## F.4 Supplementary experiments for different solution accuracies

In this section, we investigate the impact of different solution accuracies on the experimental results of problems (5) and (6). We set $\epsilon$ to be either $10^{-4}$ or $10^{-7}$ and terminate the algorithms for PB-APG and PB-APG-sc when $\|\mathbf{x}_{k+1} - \mathbf{x}_k\| \leq \epsilon$. For aPB-APG and aPB-APG-sc, we choose the corresponding $\epsilon_0$ values as 1 or $10^{-3}$. The remaining settings are the same as in Section 4.

We also plot the values of the residuals of the lower-level objective $G(\mathbf{x}_k) - G^*$ and the upper-level objective over time in Figures 5 and 6. Additionally, we also collect the total number of iterations, the lower- and upper-level objective values, and the optimal gaps of our methods in Table 6 for problems (5) and (6) with different solution accuracies.

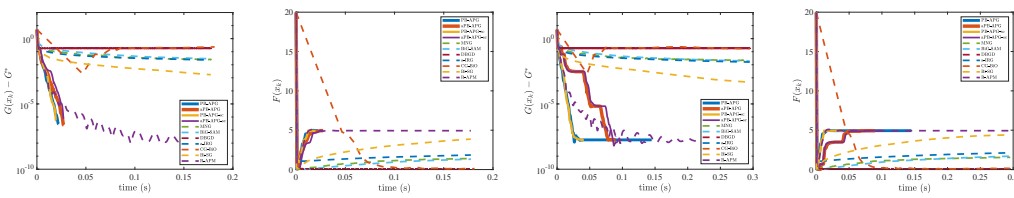

Figure 5: LRP (5) with $\epsilon = 10^{-4}$ (left two subfigures) and $\epsilon = 10^{-7}$ (right two subfigures).

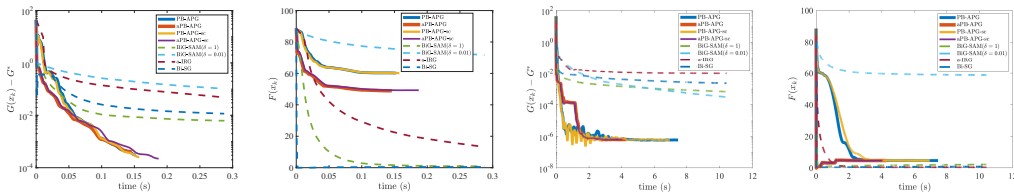

Figure 6: LSRP (6) with $\epsilon = 10^{-4}$ (left two subfigures) and $\epsilon = 10^{-7}$ (right two subfigures).

From Figures 5 and 6, it is evident that in most cases, our methods outperform the other methods in terms of both the lower- and upper-level objectives. However, there is an exception in the case of the upper-level objective for problem (6) when $\epsilon = 10^{-4}$. As illustrated in the second subfigure in Figure 6, our methods exhibit larger function values for the upper-level objective compared to the other methods (except BiG-SAM ($\delta = 0.01$)), despite still achieving smaller optimal gaps for the lower-level objective. This discrepancy actually indicates that our methods have not yet achieved the desired accuracy when $\epsilon = 10^{-4}$, and it is important to note that $\|\mathbf{x}_{k+1} - \mathbf{x}_k\| \leq \epsilon$ is not the termination criterion in our proposed algorithms, as explained in Appendix F.1. Therefore, the larger optimality gaps for the upper-level objective in this case may be attributed to the termination criterion.

Table 6: Lower- and upper-level objectives and optimal gaps with different solution accuracies for problem (5).

| $\epsilon = 10^{-4}$ | | | | | |
|---|---|---|---|---|---|
| Method | Total iterations | Lower-level value | Lower-level gap | Upper-level value | Upper-level gap |
| PB-APG | 124 | 3.2794e-01 | 2.8671e-07 | 4.9483e+00 | 6.7024e-03 |
| aPB-APG | 148 | 3.2794e-01 | 2.3660e-07 | 4.9419e+00 | 2.9831e-04 |
| PB-APG-sc | 100 | 3.2794e-01 | 5.4674e-07 | 4.9287e+00 | -1.2956e-02 |
| aPB-APG-sc | 149 | 3.2794e-01 | 7.9015e-07 | 4.9302e+00 | -1.1404e-02 |
| $\epsilon = 10^{-7}$ | | | | | |
| Method | Total iterations | Lower-level value | Lower-level gap | Upper-level value | Upper-level gap |
| PB-APG | 841 | 3.2794e-01 | 1.7631e-08 | 4.9382e+00 | -3.3999e-03 |
| aPB-APG | 551 | 3.2794e-01 | 1.7707e-08 | 4.9382e+00 | -3.4075e-03 |
| PB-APG-sc | 225 | 3.2794e-01 | 1.7493e-08 | 4.9383e+00 | -3.3691e-03 |
| aPB-APG-sc | 614 | 3.2794e-01 | 1.7507e-08 | 4.9382e+00 | -3.3874e-03 |

Table 7: Lower- and upper-level objectives and optimal gaps with different solution accuracies for problem (6).

| $\epsilon = 10^{-4}$ | | | | | |
|---|---|---|---|---|---|
| Method | Total iterations | Lower-level value | Lower-level gap | Upper-level value | Upper-level gap |
| PB-APG | 426 | 7.6950e-03 | 3.0342e-04 | 6.0249e+01 | 5.5407e+01 |
| aPB-APG | 432 | 7.8018e-03 | 4.1016e-04 | 4.8967e+01 | 4.4125e+01 |
| PB-APG-sc | 437 | 7.6456e-03 | 2.5400e-04 | 6.0196e+01 | 5.5354e+01 |
| aPB-APG-sc | 517 | 7.6143e-03 | 2.2274e-04 | 4.9292e+01 | 4.4449e+01 |
| $\epsilon = 10^{-7}$ | | | | | |
| Method | Total iterations | Lower-level value | Lower-level gap | Upper-level value | Upper-level gap |
| PB-APG | 13707 | 7.3922e-03 | 5.9756e-07 | 4.7279e+00 | -1.1460e-01 |
| aPB-APG | 7803 | 7.3923e-03 | 6.5025e-07 | 4.7300e+00 | -1.1248e-01 |
| PB-APG-sc | 12724 | 7.3922e-03 | 5.7840e-07 | 4.7354e+00 | -1.0714e-01 |
| aPB-APG-sc | 7429 | 7.3922e-03 | 6.3816e-07 | 4.7326e+00 | -1.0992e-01 |

Tables 3, 6, and 7 demonstrate that the number of iterations for our methods also increases with the solution accuracy, while the optimal gaps of the lower- and upper-level objectives decrease correspondingly. This finding confirms that the number of iterations and the optimal gaps are influenced by the solution accuracy, as illustrated in the expressions for the number of iterations provided by Theorem 3.3 and other related theorems.

