# OpenReview forum: "Penalty-based Methods for Simple Bilevel Optimization under Hölderian Error Bounds"
_NeurIPS.cc/2024/Conference — NeurIPS 2024 poster_

### Official Review · Reviewer_j2Bu · 2024-07-04

**Soundness:** 4
**Presentation:** 3
**Contribution:** 3
**Rating:** 7
**Confidence:** 3

**Summary:**

This article presents a nice extension to the broad class of penalty method problems, particularly for the case when the objective functions have a Hölderian error bound. This assumption for this class of problems appears new, and a thorough coverage of interesting results are claimed.

**Strengths:**

- New results presented for bilevel optimization for the Hölderian error bound setting
- New relationships between the minimizers of a penalty problem and the exact problem; this is a particularly mature field, and finding new results here is meritorious.
- Presentation of a standard prox-based method for solving these bilevel problems with their new theory, with complexity details explained.

**Weaknesses:**

- The article does not distinguish between what type of subdifferential is used. Since the authors also make a mention of the applicability of their results to nonconvex objectives, this is crucial point. Many classical results in optimization and convex analysis are proven using the *convex* subdifferential; however, even for differentiable nonconvex functions (e.g., $-x^2$), the convex subdifferential is empty, while other notions like the Clarke subdifferential is nonempty.
- Assumption 3.2 is a very strong one; while I agree with the author(s) that the prox of a sum is widely studied, the list of settings in which one can compute this is still quite limited. However, this is not a major drawback for me, since handling HEB bilevel optimization even in the L-smooth setting appears to be new.
- In my opinion, leaving lines 140--142 to stake claim to results for non-convex which are exclusively in the appendix seems improper. The appendix is technically "unreviewed" material; if the formal statement of the result cannot fit into this article, then I do not think it is worth including. I suggest that the authors either try to find a way to incorporate these results into the article (since they are indeed quite interesting), or save the results for another work.
- The graphs of the numeric are too small for me to read, especially in-print.

Minor comments:
- The fact that the argmin of  a convex lsc function is closed and convex is a classical result which has been known for decades, so it seems odd to cite [Beck and Sabach, 2014] instead of a classical book, or the article where it was first proven.
- The article makes mention to functions $f_2$, $g_2$ far before they are introduced, which is confusiung to the reader.
- Names "Pock, Tseng, etc." are not capitalized in the citations.
- Citation on lines 391-392 is missing volume number.

**Questions:**

The classical article on penalty methods,

Dmitri P. Bertsekas, "Necessary and Sufficient conditions for a penalty method to be exact", Math. Program., vol. 9, pp. 87 - 99, 1975.

basically states that, if the penalty function is smooth near the solution set, then the penalty must go to infinity in order to achieve exact penalization; and, if the penalty function is nonsmooth near the solution set, then a finite penalty will suffice. My question is: are your results consistent with this article? In particular,
  - In the setting where $G$ is smooth, can you confirm that $\lim_{\varepsilon\to 0}\gamma^*\to\infty$?
  - On the other hand, if $G$ is nonsmooth at its solution set, does $\lim_{\varepsilon \to 0} \gamma^*$ approach a finite value? Violating this second requirement would not contradict the aforementioned article, but it would significantly strengthen your results.

**Limitations:**

The authors do a good job of explaining the required assumptions for their results.

---

> ### Author Rebuttal · Authors · 2024-08-06
>
> We acknowledge the valuable insights you have provided for our paper.
>
> # Weakness 1:
> Thank you for pointing this out. Our definition of the subdifferential is derived from convex analysis. We use the subdifferential for a convex function $f$: $\partial f(x) = \\{ g : f(y) - f(x) \ge g^T(y - x) \\}$ (cf. [1, Section 4.2]). In our analysis, the usage of the subdifferential is limited to convex functions (see Assumption 2.1 and equation (8)). For nonconvex functions, we do not use Assumption 2.1 but instead directly assume the Lipschitz continuity of $F$ to simplify our analysis.
>
> For the general nonconvex case, we can also use other subdifferentials for 'stationary points' as discussed in Section 4 of [2], such as B(ouligand)-stationary points or C(larke)-stationary points, using the B-subdifferential or C-subdifferential, respectively. However, it is often challenging to obtain a simple definition of stationarity for the original simple bilevel optimization problem when the lower-level problem is nonconvex. Therefore, in our future work, we will explore the relationship between stationary points of the original problem and the penalized problem using general subdifferentials in scenarios where the upper-level problem is nonconvex but the lower-level problem is convex.
>
> # Weakness 2:
> Thank you for recognizing the novelty of our work. The requirement for the nonsmooth term to be prox-friendly is widely adopted in optimization and is satisfied in many machine learning applications, such as $\ell_1$- and $\ell_2$-norms. It is also important to note that our assumption is more general than those found in existing literature.
>
> Specifically, in the simple bilevel literature, when employing proximal mappings, researchers often consider the scenario where only one level contains a nonsmooth term (see, e.g., [3,4,5]). The proximal mapping of the sum $f_2 + \gamma g_2$ is then reduced to the proximal mapping of either $f_2$ or $g_2$, which is a more easily satisfied condition.
>
> A similar case occurs when $f_2 = \beta g_2$ for some $\beta > 0$. For instance, when minimizing both the validation loss and training loss of the LASSO problem simultaneously, the nonsmooth terms of the upper- and lower-level objectives are identical, both being $\ell_1$-norms. In this situation, the proximal mapping of $f_2 + \gamma g_2$ corresponds to the proximal mapping of a $\lambda \ell_1$-norm, for some $\lambda>0$, which is straightforward to compute.
>
> Generally, the prox-friendly properties of $f_2 + \gamma g_2$ are challenging to maintain. However, some studies have identified conditions under which the sum of two proximal mappings can be easily computed (e.g., [6,7]).
> For example, in regression problems, it is common to encounter situations where the nonsmooth parts of the upper- and lower-level objectives are the indicator functions of an $\ell_1$-norm ball and an $\ell_2$-norm ball, respectively. The joint proximal mapping is then the projection onto the intersection of these two balls, and the method in [8] can be used to compute it. Another case involves one level having an $\ell_2$-norm regularizer and the other having an $\ell_1$-norm regularizer. The sum $\lambda_1\\|x\\|\_2^2 + \lambda_2\\|x\\|\_1^2$ ($\lambda_1, \lambda_2 > 0$) is known as the elastic net, for which the proximal mapping has a closed form.
>
> # Weakness 3:
> Thank you for recognizing the significance of our analysis of non-convex problems. Due to space constraints, we initially placed the results in the appendix. In our revision, we will restructure the paper to include the analysis of non-convex problems in the main paper.
>
> # Weakness 4:
> We apologize for any inconvenience. In our revision, we will adjust the size and format of the graphs to improve their readability and comprehensibility. Furthermore, we have included the experimental results of our main paper in 'Figure 4' and 'Figure 5' of the PDF in 'Author Rebuttal'.
>
> - Minor comment 1: Thank you for pointing this out. The correct references should be [1, Proposition 1.2.2 and Page 49].
>
> - Minor comment 2: We apologize for the omission of the introduction of $f_2$ and $g_2$ before their usage.
>
> - Minor comments 3, and 4: Thank you for pointing out these typos. We will correct them in our revised paper.
>
> # Questions:
> Thanks for bringing Bertsekas [9] to our attention. We found that the assumptions and conditions in [9] are different from ours.
> [9] suggests a scalar penalty function $p(t): R\to R$ satisfying that $p$ is convex and $p(t)=0$ for all $t\le 0$ and $p(t)>0$ for all $t>0$. However, in our paper, we use $p(t)=\gamma \cdot t$, which contradicts $p(t)=0$ for all $t\le 0$. Nevertheless, let us compare the results.
>
> First, in Section 2, [9] requires an assumption (A.2) that problem $(P_{\text{val}})$
> $$
> \min F(x)\quad \text{s.t.} \quad G(x)-G^*\le0.
> $$
> has at least one optimal Lagrange multiplier. This assumption often fails (the multiplier is often $\infty$) in simple bilevel optimization as the Slater condition fails for $G(x)-G^*\le0$.
>
> In the case of $\alpha=1$ in the HEB assumption, the multiplier exists. For $p(t)$ satisfying conditions in [9], Proposition 1 in [9] says that a necessary condition for exact penalization is $\lim_{t\to0^+}\frac{p(t)}{t}\ge y$ and a sufficient condition is $\lim_{t\to0^+}\frac{p(t)}{t}>y$, where $y$ is an optimal Lagrange multiplier. Unless some multiplier is zero (this means  $(P_{\text{val}})$ is essentially unconstrained), the latter happens only if $p$ is nonsmooth. We guess this leads to your question. However, the smoothness is related to $p$, but not $G$.
>
> In our case, we always have  $\lim_{t\to0^+}\frac{p(t)}{t}=\gamma>0$. But as A.2 and C.2 in [9] fail here, there is no contraction between [9] and our results.
>
> These observations indicate that the results proposed by [9] don't apply to our paper.
>
> We sincerely hope that the preceding discussion addresses your inquiries. We appreciate your constructive questions once again.

---

> > ### Comment · Reviewer_j2Bu · 2024-08-13
> >
> > I thank the authors very much for their clear and thorough responses to both my questions and concerns. I am quite happy to hear that the revised version will include the precise subdifferential definition used, as well as the nonconvex results. I have decided to increase my score.

---

> > > ### Author Response · Authors · 2024-08-13
> > >
> > > We deeply appreciate your acknowledgement and decision to raise the score.

---

> ### Author Response · Authors · 2024-08-06
> **References**
>
> [1] Bertsekas, Dimitri, Angelia Nedic, and Asuman Ozdaglar. Convex analysis and optimization. Vol. 1. Athena Scientific (2003).
>
> [2] Cui Y, Liu J, Pang J S. Nonconvex and nonsmooth approaches for aﬀine chance-constrained stochastic programs[J]. Set-Valued and Variational Analysis, 2022, 30(3): 1149-1211.
>
> [3] Doron, Lior, and Shimrit Shtern. Methodology and first-order algorithms for solving nonsmooth and non-strongly convex bilevel optimization problems. Mathematical Programming 201.1 (2023): 521-558.
>
> [4] Jiang R, Abolfazli N, Mokhtari A, Hamedani EY. A conditional gradient-based method for simple bilevel optimization with convex lower-level problem. In International Conference on Artificial Intelligence and Statistics. PMLR (2023): pp. 10305-10323.
>
> [5] Merchav, Roey, and Shoham Sabach. Convex Bi-level Optimization Problems with Nonsmooth Outer Objective Function. SIAM Journal on Optimization 33.4 (2023): 3114-3142.
>
> [6] Yu, Yao-Liang. On decomposing the proximal map. Advances in neural information processing systems 26 (2013).
>
> [7] Pustelnik, Nelly, and Laurent Condat. Proximity operator of a sum of functions; application to depth map estimation. IEEE Signal Processing Letters 24.12 (2017): 1827-1831.
>
> [8] Liu, Hongying, Hao Wang, and Mengmeng Song. Projections onto the intersection of a one-norm ball or sphere and a two-norm ball or sphere. Journal of Optimization Theory and Applications 187 (2020): 520-534.
>
> [9] Bertsekas, Dimitri P. Necessary and sufficient conditions for a penalty method to be exact. Mathematical programming 9.1 (1975): 87-99.

---

### Official Review · Reviewer_fmaf · 2024-07-11

**Soundness:** 4
**Presentation:** 4
**Contribution:** 3
**Rating:** 5
**Confidence:** 3

**Summary:**

This paper deals with a simple bilevel (the lower-level problem has no dependence on the upper-level variable) optimization problem, where both the upper- and the lower-level objectives are convex and potentially non-smooth. To simplify the original bilevel problem the authors consider a penalty-based single-level reformulation and establish the relationship between the solutions of the two problems. An accelerated proximal gradient method (PB-APG)  is proposed to solve the penalty problem. Under a Holderian error bound condition and certain additional assumptions about the convexity and smoothness of the objectives, convergence of the PB-APG is shown to approximate global optimal solutions of the original bilevel problem.

**Strengths:**

* Differently from most of the works in literature, this paper deals with a problem in which the global solutions of the original bilevel problem are attainable. As a result, from a theoretical perspective, this is an interesting problem class. In addition, the proposed method is shown to (approximately) converge to these global solutions (of the original bilevel problem) and not to the solutions of some reformulation, as it is usually the case in other bilevel works.
* The authors provide a comprehensive and detailed summary of algorithms developed for solving simple bilevel optimization problems (in table 1).

**Weaknesses:**

* From an applications perspective, the problem class considered here does not seem very interesting. The bilevel problem is simple, the objectives are (strongly) convex, a Holderian error bound holds and the non-smooth term is required to be proximal-friendly. This significantly restricts the number and complexity of potential applications. Indeed, the applications presented in the appendix seem to revolve around (regularized) linear least-squares problems and simple learning models like logistic regression.
* The proposed algorithm solves the reformulated minimization problem $P_{\Phi}$ rather than the original bilevel. This minimization problem looks fairly standard, and it seems that known algorithms are applied directly for its solution. Thus, it appears that there is no significant novelty in terms of algorithm design.

**Questions:**

* Are Holderian error bounds satisfied in problems involving training of complex learning models, like neural networks.
* In table 1 some of the algorithms depend on parameters $\alpha$ and b, but the property to which these parameters correspond to is not specified. Please clarify this.

**Limitations:**

No negative societal impact

---

> ### Author Rebuttal · Authors · 2024-08-06
>
> We sincerely appreciate your valuable insights and thoughtful feedback regarding this paper.
>
> # Weakness 1:
> We should emphasize that the problem class considered here receives a lot of interest in the literature.
>
> - Simplicity: Although bilevel problems may appear straightforward, they have numerous applications and have garnered significant attention in the optimization and machine learning communities. For example, dictionary learning [1], lexicographic optimization [2], and lifelong learning [3].
>
> - Convexity: Almost all existing papers on simple bilevel optimization (SBO) investigate cases where the objectives at both levels are convex (see Section 1.1 and Table 1 of our paper). Although some practical models do not meet this assumption, they often exhibit local convexity in the vicinity of their minimizers. Additionally, we have considered the nonconvex case (refer to l. 140-142 and Appendix D).
>
> - Holderian error bound (HEB): In the literature on SBO, the HEB is a commonly utilized assumption [4,5]. In Appendix C of our paper, we demonstrate that this assumption applies to many practical problems.
>
> - Prox-friendly property: The requirement for the non-smooth term to be prox-friendly is widely adopted in optimization and is satisfied in many machine learning applications, such as $\ell_1$- and $\ell_2$-norms. It is important to note that our assumption is more general than existing literature.
>
>    In the simple bilevel literature, when employing proximal mappings, researchers often consider the scenario where only one level contains a nonsmooth term (see, e.g., [3,4,5,6]). The proximal mapping of the sum $f_2 + \gamma g_2$ is then reduced to the proximal mapping of either $f_2$ or $g_2$, which is a more easily satisfied condition.
>
>    A similar case occurs when $f_2 = \beta g_2$ for some $\beta > 0$. For instance, when minimizing both the validation loss and training loss of the LASSO problem simultaneously, the nonsmooth terms of the upper- and lower-level objectives are identical, both being $\ell_1$-norms. In this situation, the proximal mapping of $f_2 + \gamma g_2$ corresponds to the proximal mapping of a $\lambda \ell_1$-norm, for some $\lambda > 0$, which is straightforward to compute.
>
>    Generally, the prox-friendly properties of $f_2 + \gamma g_2$ are challenging to maintain. However, some studies have identified conditions under which the sum of two proximal mappings can be easily computed (e.g., [7, 8]). For example, in regression problems, it is common to encounter situations where the nonsmooth parts of the upper- and lower-level objectives are the indicator functions of an $\ell_1$-norm ball and an $\ell_2$-norm ball, respectively. The joint proximal mapping is then the projection onto the intersection of these two balls, and the method in [9] can be used to compute it. Another case involves one level having an $\ell_2$-norm regularizer and the other having an $\ell_1$-norm regularizer. The sum $\lambda_1\\|x\\|\_2^2 + \lambda_2\\|x\\|\_1$ ($\lambda_1, \lambda_2 > 0$) is known as the elastic net, for which the proximal mapping has a closed form [7, Example 5].
>
> - Applications: The applications in Appendix A are widely adopted in the literature [3,4,6,10]. Additionally, there are other applications of SBO, such as sparsity representation learning, fairness regularization, and lexicographic optimization [2]. Following your suggestion, we conducted three additional, more practical, and complex simulations. The problem settings and experimental results are detailed in the global 'Author Rebuttal'.
>
> # Weakness 2:
> Utilizing the penalization method to solve the original SBO problem is a novel approach. Among the various studies on SBO (refer to Section 1.1 and Appendix B of our paper), we are the first to employ the penalization method. While the Tikhonov regularization appears similar to our framework, its origins differ. Implementing Tikhonov regularization requires the ''slow condition'' ($\lim_{k\to\infty} \sigma_k = 0, \sum_{k=0}^{\infty}\sigma_k=+\infty$), necessitating iterative solutions for each $\sigma_k$. In contrast, our method only requires solving a single optimization problem for a given $\gamma$, whose theoretical significance is clear. Theoretically, we establish the relationship between the approximate solutions of the original bilevel problem and those of the reformulated single-level problem $P_{\Phi}$ with a specific $\gamma$. This constitutes the first theoretical result linking the original bilevel problem to the penalty problem $P_{\Phi}$ with the optimal non-asymptotic complexity result.
>
> Moreover, since $\gamma$ relies on many parameters, as demonstrated in Section 3.1.2 of our paper, determining $\gamma$ can be challenging. To address this, we propose an adaptive version of PB-APG that updates $\gamma$ dynamically and invokes PB-APG with varying $\gamma$ and solution accuracies.
>
> # Question 1:
> Table 2 of our paper provides examples and references [4,5] about the HEB, which is widely used in learning models. For instance, the piecewise maximum function and least-squares loss function can be employed to train classification and regression tasks, respectively. For problems involving the complex learning models of neural networks, although these models are generally nonconvex, they may have local convex structures and also KL (or PL) conditions, which are equivalent to the HEB in the convex case [11]. Indeed, [12, Proposition 2] demonstrates that many applications in neural networks (e.g., DNNs) satisfy the KL inequality. We will explore this in our future work.
>
> # Question 2:
> We apologize for omitting the definitions of some parameters in Table 1. For IR-CG [13], the range of $p$ is $p\in(0,1)$. For our methods, the ranges of $\alpha$ and $\beta$ are $\alpha \geq 1$ and $\beta > 0$, respectively. We will clarify this in our revision.

---

> > ### Comment · Reviewer_fmaf · 2024-08-10
> > **Comment from Reviewer fmaf**
> >
> > I would like to thank the authors for their responses. I am raising my score to 5.

---

> > > ### Author Response · Authors · 2024-08-13
> > >
> > > Thank you for acknowledging our response, and for raising the score.

---

> ### Author Response · Authors · 2024-08-06
> **References**
>
> [1] Beck, Amir, and Shoham Sabach. A first order method for finding minimal norm-like solutions of convex optimization problems. Mathematical Programming 147.1 (2014): 25-46.
>
> [2] Gong, Chengyue, and Xingchao Liu. Bi-objective trade-off with dynamic barrier gradient descent. NeurIPS 2021 (2021).
>
> [3] Jiang R, Abolfazli N, Mokhtari A, Hamedani EY. A conditional gradient-based method for simple bilevel optimization with convex lower-level problem. In International Conference on Artificial Intelligence and Statistics. PMLR (2023): pp. 10305-10323.
>
> [4] Doron, Lior, and Shimrit Shtern. Methodology and first-order algorithms for solving nonsmooth and non-strongly convex bilevel optimization problems. Mathematical Programming 201.1 (2023): 521-558.
>
> [5] Sepideh Samadi, Daniel Burbano, and Farzad Yousefian. Achieving optimal complexity guarantees for a class of bilevel convex optimization problems. arXiv preprint arXiv:2310.12247, 2023
>
> [6] Merchav, Roey, and Shoham Sabach. Convex Bi-level Optimization Problems with Nonsmooth Outer Objective Function. SIAM Journal on Optimization 33.4 (2023): 3114-3142.
>
> [7] Yu, Yao-Liang. On decomposing the proximal map. Advances in neural information processing systems 26 (2013).
>
> [8] Pustelnik, Nelly, and Laurent Condat. Proximity operator of a sum of functions; application to depth map estimation. IEEE Signal Processing Letters 24.12 (2017): 1827-1831.
>
> [9] Liu, Hongying, Hao Wang, and Mengmeng Song. Projections onto the intersection of a one-norm ball or sphere and a two-norm ball or sphere. Journal of Optimization Theory and Applications 187 (2020): 520-534.
>
>
> [10] Mostafa Amini and Farzad Yousefian. An iterative regularized incremental projected subgradient method for a class of bilevel optimization problems. In 2019 American Control Conference (ACC), pages 4069–4074. IEEE, 2019.
>
> [11] Bolte, Jerome, et al. From error bounds to the complexity of first-order descent methods for convex functions. Mathematical Programming 165 (2017): 471-507.
>
> [12] Zeng, Jinshan, et al. Global convergence of block coordinate descent in deep learning. International conference on machine learning. PMLR, 2019.
>
> [13] Khanh-Hung Giang-Tran, Nam Ho-Nguyen, and Dabeen Lee. Projection-free methods for solving convex bilevel optimization problems. arXiv 2023.

---

### Official Review · Reviewer_EYqV · 2024-07-13

**Soundness:** 3
**Presentation:** 3
**Contribution:** 2
**Rating:** 5
**Confidence:** 3

**Summary:**

This work proposes a penalty based algorithm for simple bilevel optimization problems. The paper studies the relationship between the solutions of the penalized problem and the original bilevel problem. It extends the existing results on general bilevel optimization problem, which are established under PL condition, to the more generic Hölderian error bounds. The results are also extended to non-smooth upper and lower-level functions.

**Strengths:**

1) The paper is well written and easy to follow.
2) The generalization of the existing results on PL condition to Hölderian error bound is interesting.
3) The extended results on non-smooth objectives are good contributions.

**Weaknesses:**

1. Missing literature comparison/review:  The method proposed in this paper is a penalty-based method. There is a large body of work for penalty-based bilevel optimization methods. While in the related work section in the main text, there is no detailed review/comparison of the previous penalty-based bilevel optimization algorithms. It might be beneficial if the authors could discuss about the advantage of the proposed method compared with the existing ones.

2. In the current state of the paper, the experiments seem to be purely synthetic and might be a little too toy.  Though the paper is majorly a theoretical one, it would benefit to have more practical experiments. It is also of great importance to find meaningful applications for the simple bilevel optimization problem.

3. The method proposed in this work falls into the penalty-based methods. It might be interesting if the authors include a comparison with existing penalty-based bilevel optimization methods in, e.g.,

**Questions:**

1. The proposed method is for convex simple bilevel optimization problems, which might be restrictive. Can the author explain how the algorithm can be extended to the non-convex case?

2. How does this algorithm compare to the penalty-based algorithms for general bilevel optimization problems? Are there any advantages demonstrated by this algorithm as compared to the generic ones?

**Limitations:**

no potential social impact

---

> ### Author Rebuttal · Authors · 2024-08-06
>
> Your expert insights are crucial to our research, and we greatly appreciate your guidance and suggestions.
>
> # Weaknesses 1:
> We should point out that for simple bilevel optimization (SBO), there is no existing penalty method, although there exists a closed related method that is based on Tikhonov regularization (please refer to l.44-65 in our paper). Your point is correct for general bilevel optimization methods, as referenced in [1, 2, 3]. However, the general bilevel optimization model, with the form $\min\ F(x,y)$ $\text{s.t.}$ $x \in \text{argmin}_x G(x,y)$ is different from SBO as the previous one is general nonconvex, while the latter is convex. Indeed, our penalty method is partially motivated by the penalty method for general bilevel optimization in [1], but our method uses many properties from the structure of SBO. Nonetheless, following your suggestion, we will include a comparison of our method and other existing methods on general bilevel penalty-based methods in our revised paper.
>
> # Weaknesses 2:
> Thanks for the comment. In fact, these experiments are widely adopted in the SBO literature [10, 11, 12, 13]. It would be more persuasive to add more practical experiments. Following your suggestion, we conduct three more practical and complex simulations. The problem settings and experimental results are presented in the global 'Author Rebuttal'. Our experimental results show that our methods exhibit faster convergence rates, and achieve the smallest optimal gaps compared to the baseline methods for both upper- and lower-level objectives. This finding confirms our theoretical predictions in the context of this practical simulation. Moreover, the third experimental results (fair classification problem) also suggest that our methods may perform well even in non-convex cases.
>
> # Weaknesses 3:
> Please refer to the answer to Weakness 1.
>
> # Question 1:
> We actually considered the nonconvex case in l.140-142 of our main paper. However, due to the page limit, we put our discussions in Appendix D.1 of our paper (please refer to "full\_paper.pdf" in the supplementary zip), we extend the penalization framework to the setting where $F$ is non-convex with the assumption of the Lipschitz (or local Lipschitz) continuous of $F$.
>
> # Question 2:
> - For a comparison with penalty-based methods for general bilevel optimization problems, please refer to 'Weaknesses 1'. Our method explores the structure of SBO and specifies the penalty parameter \\(\gamma\\) to establish a relationship between the original SBO and the penalty formulation.
>
> - For general bilevel optimization problems, there have been recent results on convergent guarantees [1,14,15, 16,17]. Among those, the one that is the most related to ours is [1]. [1] investigates the case when the upper-level objective is nonconvex and gives convergence results under additional assumptions [1, Theorem 3 and 4].  However, as the general bilevel optimization problem is nonconvex, the algorithms in the literature often converge to weak stationary points, while our method for SBO converges to global optimal solution.

---

> ### Author Response · Authors · 2024-08-06
> **References**
>
> [1] Han Shen and Tianyi Chen. On penalty-based bilevel gradient descent method. ICML 2023.
>
> [2] Lu, Zhaosong, and Sanyou Mei. First-order penalty methods for bilevel optimization. SIAM JOPT 2024.
>
> [3] Kwon, Jeongyeol, Dohyun Kwon, Steve Wright, and Robert Nowak. On penalty methods for nonconvex bilevel optimization and first-order stochastic approximation. arxiv 2023.
>
> [4] Phillips, David L. A technique for the numerical solution of certain integral equations of the first kind. JACM 1962.
>
> [5] Doron, Lior, and Shimrit Shtern. Methodology and first-order algorithms for solving nonsmooth and non-strongly convex bilevel optimization problems. MP 2023.
>
> [6] Jiang, R., Abolfazli, N., Mokhtari, A. and Hamedani, E.Y. A conditional gradient-based method for simple bilevel optimization with convex lower-level problem. AISTATS 2023.
>
> [7] Gong C, Liu X. Bi-objective trade-off with dynamic barrier gradient descent[J]. NeurIPS 2021, 2021.
>
> [8] Cao J, Jiang R, Abolfazli N, et al. Projection-free methods for stochastic simple bilevel optimization with convex lower-level problem[J]. Advances in Neural Information Processing Systems, 2024, 36.
>
> [9] Cui Y, Liu J, Pang J S. Nonconvex and nonsmooth approaches for affine chance-constrained stochastic programs[J]. Set-Valued and Variational Analysis, 2022, 30(3): 1149-1211.
>
> [10] Mostafa Amini and Farzad Yousefian. An iterative regularized incremental projected subgradient method for a class of bilevel optimization problems. In 2019 American Control Conference (ACC), pages 4069–4074. IEEE, 2019.
>
> [11] Ruichen Jiang, Nazanin Abolfazli, Aryan Mokhtari, and Erfan Yazdandoost Hamedani. A condi- tional gradient-based method for simple bilevel optimization with convex lower-level problem. In International Conference on Artificial Intelligence and Statistics, pages 10305–10323. PMLR, 2023.
>
> [12] Sepideh Samadi, Daniel Burbano, and Farzad Yousefian. Achieving optimal complexity guarantees for a class of bilevel convex optimization problems. arXiv preprint arXiv:2310.12247, 2023.
>
> [13] Lingqing Shen, Nam Ho-Nguyen, and Fatma Kılınç-Karzan. An online convex optimization-based framework for convex bilevel optimization. Mathematical Programming, 198(2):1519–1582, 2023.
>
> [14] Risheng Liu, Yaohua Liu, Shangzhi Zeng, and Jin Zhang. Towards gradient-based bilevel opti- mization with non-convex followers and beyond. Advances in Neural Information Processing Systems, 34:8662–8675, 2021.
>
> [15] Daouda Sow, Kaiyi Ji, Ziwei Guan, and Yingbin Liang. A primal-dual approach to bilevel optimization with multiple inner minima. arXiv preprint arXiv:2203.01123, 2022.
>
> [16] Lesi Chen, Jing Xu, and Jingzhao Zhang. On bilevel optimization without lower-level strong convexity. arXiv preprint arXiv:2301.00712, 2023.
>
> [17] Feihu Huang. On momentum-based gradient methods for bilevel optimization with nonconvex lower-level. arXiv preprint arXiv:2303.03944, 2023.

---

### Official Review · Reviewer_WQ6y · 2024-07-15

**Soundness:** 3
**Presentation:** 3
**Contribution:** 2
**Rating:** 7
**Confidence:** 3

**Summary:**

The authors propose algorithms to solve simple bilevel optimization problems of the form $\min_x F(x)$ s.t. $x$ minimizes $G$, in the case where $F$ and $G$ are composite convex functions (i.e. some of 2 convex functions, one of which is also smooth), and assuming Lipschitz continuity of $F$ on the set of minimizers of $G$ and $G$ verifying a Holder error bound.

The proposed algorithms actually tackle a simple minimization problem, solving a relaxed version of the original problem that authors previously proved to be equivalent.

**Strengths:**

The link between the solutions of the original problem and the ones of the relaxed one is interesting and allows considering a simpler problem to tackle.

**Weaknesses:**

I am not very familiar with this specific piece of literature, so I cannot pretend to be sure of the novelty of the results.
In particular, in line 47, the authors say « Tikhonov regularization most related to $P_\gamma$ », I do not see any difference between the two formulations. Can the authors develop on this? What is exactly done in Tikhonov's paper? And what is new here?


Minors:
- Usually, a proximal-based algorithm can be applied to the particular case where the non-smooth part of the problem is null, then the prox map is simply the identity map, and all the guarantees hold. But in this paper's setting, the non-smooth part is essential and cannot be considered null as a particular case. Indeed, otherwise, $G$ would be smooth, therefore upper bounded by a quadratic function, and $G$ is also assumed to grow faster than $||x-\bar{x}||^{alpha}$ which cannot happen (if alpha<2).
- A natural oracle assumption in the case where $F$ and $G$ contain each a non-smooth part would be to assume access to the proximal map associated with each. However, authors assume access to the proximal of all the linear combinations of the 2 non-smooth parts, and this choice seems to be only motivated by the reformulation of the problem. Doing this makes the derivation of PB-APG trivial once Thm 2.5. has been proven. But how could we tackle the same problem with the sole knowledge of the 2 prox maps independently?

**Questions:**

- l.32: Why exclude the particular case where $X_{opt}$ is a singleton? This assumption is not used, therefore if we do not know in advance whether $X_{opt}$ is a singleton or not, one can still run this paper's algorithms, and guarantees will follow.
- l.39: Even if what $G^*$ is seems clear, I think it is worth mentioning it before using it in line 35.
- l.50: What about the upper problem?
- l.50: « where b in (0, 0.5) ». The authors should be more specific here. Is it true for one specific b in the interval, depending on some parameters of the problem? Is it true for any (one proposed algorithm for each value of b)?
- l.55: same remark
- l.61: It would be more convenient if the authors keep a consistent way of displaying complexity.
- l.68: What is g_2? Not defined yet at this stage. Moreover, note that the authors could not look at what happens in their case when $g_2=0$, as discussed in the "weakness" section.
- l.101: What is f_2? Again not defined. And I think here that the authors meant $F$.
- l.193: typo: « an » is repeated.
- l.205: « while samadi … » If they consider the smooth case only, studying $\alpha<2$ would be pointless because it is empty.
- It would be great if authors could add explicit stopping criteria in their algorithms. Indeed, if $F^*$ and $G^*$ are not known, I guess we stop PB-APG after K iterations where K is set according to Th3.3. But then, PB-APG is not only a function of $\phi$, $\psi$, $L_{f_1}$, $L_{g_1}, x_0, \epsilon$, but also $\ell_F$, $\alpha$ and $\beta$. Which is important to actually understand the iterates of aPB-APG.

**Limitations:**

No limitation

---

> ### Author Rebuttal · Authors · 2024-08-05
>
> Thanks for providing these valuable suggestions.
> # Weakness:
> Solodov [1] applied Tikhonov regularization (TR) [2] to solve simple bilevel optimization problems. Although the formulation of TR (l.47-48 in our paper) is similar to our method (l.36-37 in our paper), their origins and theories differ. Implementing TR requires the "slow condition" ($\lim_{k\to\infty} \sigma_k=0,\sum_{k=0}^{\infty}\sigma_k=+\infty$) [2], while we do not have this constraint; Our method provides non-asymptotic convergence rates, whereas TR only provides asymptotic results on either the upper-level or the lower-level problem (l.537 in Appendix B); We give an explanation on $\gamma$ while they do not.
> - Minor weakness 1: Thank you for pointing this out. The $\alpha$ in the error bound assumption is not arbitrary. Indeed, it is determined by the lower-level problem. When there exists some $\alpha$ such that $\text{dist}(x,X_{\text{opt}})^\alpha\le\rho p(x)$, the subsequent analysis is correct. For your example, suppose the function $G$ is $L$-smooth and $x^*$ is unique, we have $G(x)-G^*\le \frac{L}{2}\\|x-x^*\\|^2$. In this case, there is no contradiction if $G$ enjoys quadratic growth ($\alpha=2$), i.e., there exists $\mu>0$ such that $G(x)-G^*\ge\frac{\mu}{2}\\|x-x^*\\|^2$. But in the L-smooth case, we cannot find some $\alpha<2$ satisfying our assumption as you observed.
> - Minor weakness 2: You are right that Assumption 2.2 is strong in some scenarios. However, we should emphasize that it is more general than existing literature. In the simple bilevel problem, when using proximal mappings, people often only consider specific cases. E.g., [3] explores 'norm-like' upper functions, [4,5] require both $F$ and $G$ to be smooth, and [6] assumes that the upper-level objective is smooth and strongly convex. Additionally, although [7] assumes the nonsmooth terms of $F$ and $G$ are prox-friendly, respectively, their complexity, $O(\max\\{1/\epsilon_F^{\frac{1}{1-a}},1/\epsilon_G^{\frac{1}{a}}\\})$ with $a\in(0.5,1)$, is significantly worse than ours.
>
>    Furthermore, in practice, the sum of proximal mapping can be reduced to the proximal mapping of either $f_2$ or $g_2$. One example is when $f_2=\beta g_2$, for some $\beta>0$. E.g., when we want to minimize the validation loss and training loss of the LASSO problem simultaneously, the non-smooth terms of the upper- and lower-level objectives are identical, both of which are $\ell_1$-norms. Then the proximal mapping of $f_2+\gamma g_2$ is the same as the proximal mapping of $\lambda\ell_1$-norm, for some $\lambda>0$, which is easy to obtain.
>
>    In addition, there also exist studies that investigate the sum of two proximal mappings ([8]: decompose it into the sum of individual proximal maps; [9]: compute the sum of proximal mappings efficiently). For example, when we are dealing with the regression problem, we often come into a situation where the nonsmooth part of the upper- and lower-level objectives are the indicator functions of an $\ell_1$-norm ball and an $\ell_2$-norm ball, respectively. The joint proximal mapping is the projection onto the intersection of these two balls. [10] shows how to compute it. Another case is that one level has a squared $\ell_2$ norm regularizer and the other has a $\ell_1$-norm regularizer. The sum of $\lambda_1\\|x\\|_2^2+\lambda_2\\|x\\|_1$ ($\lambda_1,\lambda_2>0$) is known as the elastic net, where the proximal mapping admits a closed form [8, Example 5].
> # Questions:
> - l.32: Yes, you are correct.  The algorithm presented in our paper can still be executed without knowing whether $ X_{\text{opt}}$ is a singleton. We will fix this in our revision.
> - l.39: Sorry for the confusion. We will introduce $G^*$ before using it in our revised version.
> - l.50: [11] guarantees asymptotic convergence for the upper-level problem as shown in Appendix B in our paper. We will clarify this in our revision. For the parameter $b\in(0,0.5)$, it does not depend on any parameter of the problem and is arbitrary in $(0,0.5)$, as stated in [11, Theorem 2]. Thank you for pointing out the ambiguity.
> - l.55: According to [12, Theorem 3.3 and Corollary 3.4], parameter $b$ is also arbitrary in $(0,0.5)$. It is not related to any parameter of functions but is instead related to the adaptive choice of step size $\eta_k$ in Algorithm 2.1 and 3.1, where $\eta_k=\frac{\eta_0}{(k+1)^b}$.
> - l.61 Thanks for your suggestion. In l.61, the convergence rate of '$O(\epsilon^{-0.5})$' is equivalent to '$O(1/K^{2})$' if we use the total number of iterations $K$. We will modify it in our revised paper.
> - l.68: Sorry. We should define $g_2$ before its first use. Or the sentence "when $F$ is strongly convex and $g_2 = 0$" should be modified to "when $F$ is strongly convex and $G$ is smooth". Moreover, for $g_2=0$, please refer to Minor weakness 1.
> - l.101 and 193: Yes, we mean $F$ instead of $f_2$. Thanks for pointing out these typos.
> - l.205: If they consider the smooth case only, it is still possible for $g_1$ to satisfy the error bound with $\alpha \ge 2$; please refer to Minor weakness 1.
> - Thanks for your constructive suggestions to our algorithms. For the stopping criterion and $K$:
>     - In Algorithm 1, we can use the following stopping criterion: we stop the loop of l.3-4 if $\frac{2(L_f+\gamma L_g)R^2}{(k+1)^2} \le\epsilon$, where $R$ is such that $\\|x_0-x^*\\|\le R$.
>     - In Thm3.3, the complete expression of $K$ is
>      $K=\sqrt{\frac{2(L_{f_1}+\gamma L_{g_1})}{\epsilon}}\\|x_0-x^*\\|-1$.
>     - In both the above criterion and $K$, parameter $\gamma$ is chosen as:
>         - If $\alpha>1$, $ \gamma=\rho l_F^{\alpha}(\alpha-1)^{\alpha-1}\alpha^{-\alpha}\epsilon^{1-\alpha}+2l_F^{\beta}\epsilon^{1-\beta}$;
>         - If $\alpha=1$, $ \gamma=\rho l_F+l_F^{\beta}\epsilon ^{1-\beta}$.
>     - In Algorithms 2,3,4, and the corresponding theorems, we can also provide criteria and the value of $K$. We will include these in the revision.

---

> > ### Comment · Reviewer_WQ6y · 2024-08-13
> > **Response to authors' rebuttal**
> >
> > I thank the authors for responding to my minor concerns.
> > Overall, the paper is well-written, sound and, I think, interesting for the Neurips community.
> > If the authors process the discussed modifications/clarifications, then I am in favor of the acceptance of this paper.

---

> > > ### Author Response · Authors · 2024-08-13
> > >
> > > Thank you for acknowledging our paper and response, and for the positive score.

---

> ### Author Response · Authors · 2024-08-06
> **References**
>
> [1] Mikhail Solodov. An explicit descent method for bilevel convex optimization. Journal of Convex Analysis, 14(2):227–237, 2007.
>
> [2] Andre Nikolaevich Tikhonov and V. I. A. K. Arsenin. Solutions of ill-posed problems. Wiley, 1977.
>
> [3] Doron, Lior, and Shimrit Shtern. Methodology and first-order algorithms for solving nonsmooth and non-strongly convex bilevel optimization problems. Mathematical Programming 201.1 (2023): 521-558.
>
> [4] Jiang R, Abolfazli N, Mokhtari A, Hamedani EY. A conditional gradient-based method for simple bilevel optimization with convex lower-level problem. In International Conference on Artificial Intelligence and Statistics. PMLR (2023): pp. 10305-10323.
>
> [5] Khanh-Hung Giang-Tran, Nam Ho-Nguyen, and Dabeen Lee. Projection-free methods for solving convex bilevel optimization problems. arXiv preprint arXiv:2311.09738, 2023.
>
> [6] Sabach, Shoham, and Shimrit Shtern. A first order method for solving convex bilevel optimization problems. SIAM Journal on Optimization 27.2 (2017): 640-660.
>
> [7] Merchav, Roey, and Shoham Sabach. Convex Bi-level Optimization Problems with Nonsmooth Outer Objective Function. SIAM Journal on Optimization 33.4 (2023): 3114-3142.
>
> [8] Yu, Yao-Liang. On decomposing the proximal map. Advances in neural information processing systems 26 (2013).
>
> [9] Pustelnik, Nelly, and Laurent Condat. Proximity operator of a sum of functions; application to depth map estimation. IEEE Signal Processing Letters 24.12 (2017): 1827-1831.
>
> [10] Liu, Hongying, Hao Wang, and Mengmeng Song. Projections onto the intersection of a one-norm ball or sphere and a two-norm ball or sphere. Journal of Optimization Theory and Applications 187 (2020): 520-534.
>
> [11] Mostafa Amini and Farzad Yousefian. An iterative regularized incremental projected subgradient method for a class of bilevel optimization problems. In 2019 American Control Conference (ACC), pages 4069–4074. IEEE, 2019.
>
> [12] Kaushik, Harshal D., and Farzad Yousefian. A method with convergence rates for optimization problems with variational inequality constraints. SIAM Journal on Optimization 31.3 (2021): 2171-2198.

---

### Author Rebuttal · Authors · 2024-08-06

In this global 'Author Rebuttal', we upload our additional experimental results, as well as the clearer and more readable experimental results of Sections 4.1 and 4.2 in our paper.

# Linear regression problem
The first experiment is the sparse linear regression problem on the data ($3,000$ instances of 'YearPredictionMSD') same as the one used in Section 4.2 in our paper. We allocate $60\\%$ of the data as the training set $(A_{tr},b_{tr})$, $20\\%$ of the data as the validation set $(A_{val},b_{val})$, and the rest as the test set $(A_{test},b_{test})$ with $\frac{1}{2}\\|A_{test}x - b_{test}\\|^2$ as the test error. The sparse linear regression problem has the following form:
$$
\min\_{x}\frac{1}{2}\\|A\_{val}x-b\_{val}\\|^2 \quad \text{s.t.}\quad x\in\text{argmin}\_z ~\frac{1}{2}\\|A_{tr}z-b_{tr}\\|^2+\\|z\\|\_1.
$$

For the experimental results, please refer to 'Figure 1' in the PDF.

# Integral equation problem
In the second experiment, we explore the regularization impact of the minimal norm solution on ill-conditioned inverse problems arising from the discretization of Fredholm integral equations of the first kind [1], i.e., the solution of the integral equation problem. Following the same setting of [2], we solve the following problem:
$$
\min\_{x}x^T Q x\quad \text{s.t.}\quad x\in\text{argmin}\_{z\ge0}\frac{1}{2}\\|Az-b\\|^2,
$$
where $[A,b_T,x_T] = phillips(100)$ and $b = b_T + 0.2w$ with $w$ is sampled from a standard normal distribution, $Q = L^T L+I$ with $L=get\_l(100)$ and $I$ is the identity matrix. The functions '$phillips$' and '$get\_l$' follow from the ''regularization tools'' MATLAB package [3].

For the experimental results, please refer to 'Figure 2' in the PDF.

# Fair classification
Moreover, in the third experiment, we address the fair classification problem as described in [4], which features a non-convex upper-level objective. We randomly sample $3,000$ instances as the training set and $2,000$ as the test set. The domain of the lower-level objective is set to be $\{x:\\|x\\|_1\le10\}$.

For the experimental results, please refer to 'Figure 3' in the PDF. Given the non-convex nature of the upper-level objective, this experiment demonstrates that our algorithm can effectively handle some practical non-convex scenarios.

# Experimental results of our main paper
Here, we present the experimental results of our main paper (Sections 4.1 and 4.2) after resizing and formatting. For the detailed results of Sections 4.1 (Logistic regression problem (LRP)) and 4.2 (Least squares regression problem (LSRP)), please refer to 'Figure 4' and 'Figure 5' in the PDF, respectively.

# References

[1] Phillips, David L. A technique for the numerical solution of certain integral equations of the first kind. JACM 1962.

[2] Doron, Lior, and Shimrit Shtern. Methodology and first-order algorithms for solving nonsmooth and non-strongly convex bilevel optimization problems. MP 2023.

[3] Hansen, Per Christian. Regularization tools version 4.0 for Matlab 7.3. Numerical algorithms 46 (2007): 189-194.

[4] Jiang, R., Abolfazli, N., Mokhtari, A. and Hamedani, E.Y. A conditional gradient-based method for simple bilevel optimization with convex lower-level problem. AISTATS 2023.

---

### Decision · Program_Chairs · 2024-09-25

**Decision:**

Accept (poster)

**Comment:**

I would like to thank both the authors and reviewers for the good discussion. The reviewers agree that the paper is interesting although two reviewers are a little hesitant only recommending "borderline accept". Having looked at the paper and read the reviews, I recommend acceptance, however I urge the authors to take the reviewers feedback into account when preparing the camera-ready version.